# Lapsyn controls branch extension and positioning of astrocyte-like glia in the *Drosophila* optic lobe

Benjamin Richier[1,2], Cristina de Miguel Vijandi [1], Stefanie Mackensen[1,3] & Iris Salecker [1]

Astrocytes have diverse, remarkably complex shapes in different brain regions. Their branches closely associate with neurons. Despite the importance of this heterogeneous glial cell type for brain development and function, the molecular cues controlling astrocyte branch morphogenesis and positioning during neural circuit assembly remain largely unknown. We found that in the *Drosophila* visual system, astrocyte-like medulla neuropil glia (mng) variants acquire stereotypic morphologies with columnar and layered branching patterns in a stepwise fashion from mid-metamorphosis onwards. Using knockdown and loss-of-function analyses, we uncovered a previously unrecognized role for the transmembrane leucine-rich repeat protein Lapsyn in regulating mng development. *lapsyn* is expressed in mng and cell-autonomously required for branch extension into the synaptic neuropil and anchoring of cell bodies at the neuropil border. Lapsyn works in concert with the fibroblast growth factor (FGF) pathway to promote branch morphogenesis, while correct positioning is essential for mng survival mediated by gliotrophic FGF signaling.

[1] The Francis Crick Institute, Visual Circuit Assembly Laboratory, 1 Midland Road, London NW1 1AT, UK. [2] Present address: The Wellcome Trust/Cancer Research UK Gurdon Institute, University of Cambridge, Tennis Court Road, Cambridge CB2 1QN, UK. [3] Present address: University of Münster, Institute of Neuro- and Behavioral Biology, Badestr. 9, 48149 Muenster, Germany. Benjamin Richier and Cristina de Miguel Vijandi contributed equally to this work. Correspondence and requests for materials should be addressed to I.S. (email: iris.salecker@crick.ac.uk)

The architecture of brain regions is defined by the connectivity of diverse neurons and their partnership with glia. Astrocytes actively shape neural circuits at short and long range by influencing their wiring and function[1]. For instance, they locally regulate sensorimotor circuit formation by expressing guidance cues in the spinal cord[2]. Moreover, astrocytes promote and remove synapses[3–5], and engulf axon fragments during developmental pruning[6–8]. In addition to providing tensile strength and metabolic support to neurons, astrocytes influence neuronal activity by regulating extracellular ion and neuro-transmitter concentrations at synapses[9, 10]. To fulfill these roles, their processes closely associate with neuronal cell bodies, dendritic and axonal arbors and synaptic contacts[10, 11]. Vertebrate astroglia display a considerable regional heterogeneity[1, 12, 13]. While protoplasmic astrocytes in the cortex are star-shaped, Müller glia in the retina or Bergmann glia in the cerebellum have elongated morphologies[14]. Astrocyte branches can be sparse or highly ramified, resembling filopodia or fine veil-like sheaths[14]. However, the question as to how astrocytes acquire their characteristic shapes at specific locations has so far received little attention.

Like their vertebrate counterparts, glia in the central nervous system (CNS) of *Drosophila melanogaster* comprise several populations[10]. Surface-associated glia, consisting of perineurial and subperineurial glia, cover the entire CNS and form the blood-brain barrier. Cortex glia extend thin processes around individual or groups of neuronal somata[15]. Finally, neuropil-associated glia include ensheathing and astrocyte-like glia[16, 17]. The former cover neuropil surfaces and compartments, whereas the latter share morphological and molecular similarities with mammalian protoplasmic astrocytes[16]. They extend highly ramified processes deep into synaptic neuropils, express transporters for neurotransmitter clearance, and share core electrophysiological features with vertebrate astrocytes[5, 10, 18–22].

The fibroblast growth factor (FGF) signaling pathway plays a remarkably conserved role in controlling glial development. For instance, FGF8/FGF receptor 3 signaling promotes the elaboration of complex branches in developing murine astrocytes[23]. In the postembryonic fly CNS, FGF signaling is required for generating perineurial and cortex glia[24]. In the eye-antennal imaginal disc, this pathway controls glial proliferation, migration, differentiation, and axonal wrapping[25]. Furthermore, astrocytes in the larval ventral nerve cord (VNC) require activation of the FGF receptor Heartless (Htl) to extend processes into the neuropil[26]. Considering the pleiotropic functions of FGF signaling, other mechanisms are expected to collaborate with this pathway in controlling astrocyte branch morphogenesis. As with neurons, the acquisition of complex branching patterns by glia may require the action of so far elusive cell surface molecules.

In the *Drosophila* visual system, photoreceptors (R-cells, R1-R8) in the retina extend axons into the optic lobe, consisting of the lamina, medulla, lobula plate, and lobula[27] (Fig. 1a). Consistent with their function in visual information processing, these ganglia are structured into reiterated columnar units. The medulla, lobula plate, and lobula neuropils are additionally organized into parallel synaptic layers. Throughout development and in adults, R-cell axons and higher-order neurons closely associate with different glial cell subtypes[28, 29]. In the VNC and antennal lobe, astrocyte-like glia adopt asymmetric stellate shapes[16, 26]. However in the visual system, we and others previously described an astrocyte-like subtype among medulla neuropil glia (mng) with a columnar branching pattern[29–32].

In this study, we followed the development of astrocyte-like mng from birth to adulthood. This glial subtype is initially highly motile and acquires its final branching pattern in a stepwise manner. To gain insights into the molecular mechanisms controlling mng development, we conducted a RNA interference (RNAi) screen for secreted and cell surface proteins. Genetic analyses revealed that the transmembrane molecule leucine-rich repeat activity-regulated protein at synapses (Lapsyn)[33] is expressed in glia and essential for branch extension and anchorage of mng at the neuropil border. Branch formation required both Lapsyn and Htl FGF receptor activation, whereas Lapsyn-mediated positioning of mng at the neuropil border ensured exposure to gliotrophic FGF signals and consequently their survival. Finally, we provide evidence that *lapsyn* controls branch morphogenesis of astrocytes in other CNS regions.

## Results

**Four astrocyte-like glial variants in the medulla.** To visualize mng in the *Drosophila* visual system, we tested several Gal4 insertions for specific activity (Fig. 1b–c and Supplementary Fig. 1a–d). mng were identified by co-expression of membrane-bound green fluorescent protein (GFP) and the glial-specific homeodomain transcription factor reversed polarity (Repo)[34], and cell body position at the medulla neuropil border. *R56F03-Gal4* and *NP6520-Gal4*, markers for ensheathing glia[17, 32, 35, 36], labeled a mng subset that extended sparsely branched processes into the neuropil (Fig. 1b and Supplementary Fig. 1b). By contrast, a 2.2 kb-enhancer fragment of the locomotion defects (*loco*) gene upstream of Gal4 (*loco*$^{1.3D2}$-*Gal4*)[37] drove strong expression in astrocyte-like mng, whose processes abundantly infiltrated the neuropil layers M1-M10 (Fig. 1c). Finally we combined *R56F03-Gal4*, *loco*$^{1.3}$-*lexA,UAS-FB1.1B*$^{260b}$ and *lexAop-myrmCherry* transgenes with immunolabeling using antibodies against Repo or the homeodomain transcription factor Prospero, a marker for astrocyte-like glia in the VNC[36] (Supplementary Fig. 1g, h). This confirmed that as their VNC counterparts, astrocyte-like mng express Prospero and that above drivers label closely associated, yet genetically and morphologically distinct glial subtypes.

To image astrocyte-like mng with single cell resolution, we next combined *loco*$^{1.3D2}$-*Gal4* with the Flybow*UAS-FB1.1* transgene[31] (Fig. 1d–n). This uncovered four astrocyte-like mng variants: short and long distal mng (dmng), whose processes infiltrated neuropil layers M1-M5 and M1-M8, respectively (Fig. 1d, g–j); proximal mng (pmng or chandelier glia[29]) mainly covering layers M9 and M10 (Fig. 1e); and lateral mng (lmng) with tangential processes primarily extending in layers M7 and M8 (Fig. 1f). dmng and pmng displayed stereotypic morphologies, with a small number of main branches giving rise to a rich meshwork of secondary processes (Fig. 1g–j). dmng contributed branches to 6–10 columns ($n = 8$ for layer M5; Fig. 1k). Similar to other astrocytes[26, 38], the processes of astrocyte-like mng showed a tiled organization, but their density and distribution varied between synaptic layers, and peripheral branches from several neighboring cells converged into single columns where they slightly intermingled (Fig. 1g-j, l–n).

Further probing the molecular profile of astrocyte-like mng (Supplementary Fig. 2), we observed expression of the GABA transporter (GAT)[26], the high-affinity glutamate transporter dEAAT1[18, 19, 36] and the glutamate recycling enzyme glutamate synthase 2 (Gs2)[39]. Thus, despite their distinct morphologies, astrocyte-like glia in the medulla are functionally related to their counterparts in other CNS regions.

**Key steps of astrocyte-like mng development.** R-cells and target neurons are primarily generated during larval development. In the optic lobe, medial neuroepithelial cells in the outer proliferation center (OPC) sequentially transform into neuroblasts (Fig. 2a). These transition through five temporal identity transcription factors, producing a string of neurons directed toward

the medulla neuropil. The oldest neuroblasts transform into neuroglioblasts, switching to the production of glia during their final division[40, 41]. Combining mosaic analysis with a repressible cell marker (MARCM)[42] and Repo labeling, we confirmed that

mng originate from OPC neuroglioblasts and migrate to and stop at the most anterior cortex-neuropil interface (Fig. 2b–e). Individual clones on average gave rise to 1.56 mng ($\pm 0.7$ 95% confidence interval, $n = 9$). Quantification of phosphoHistone 3

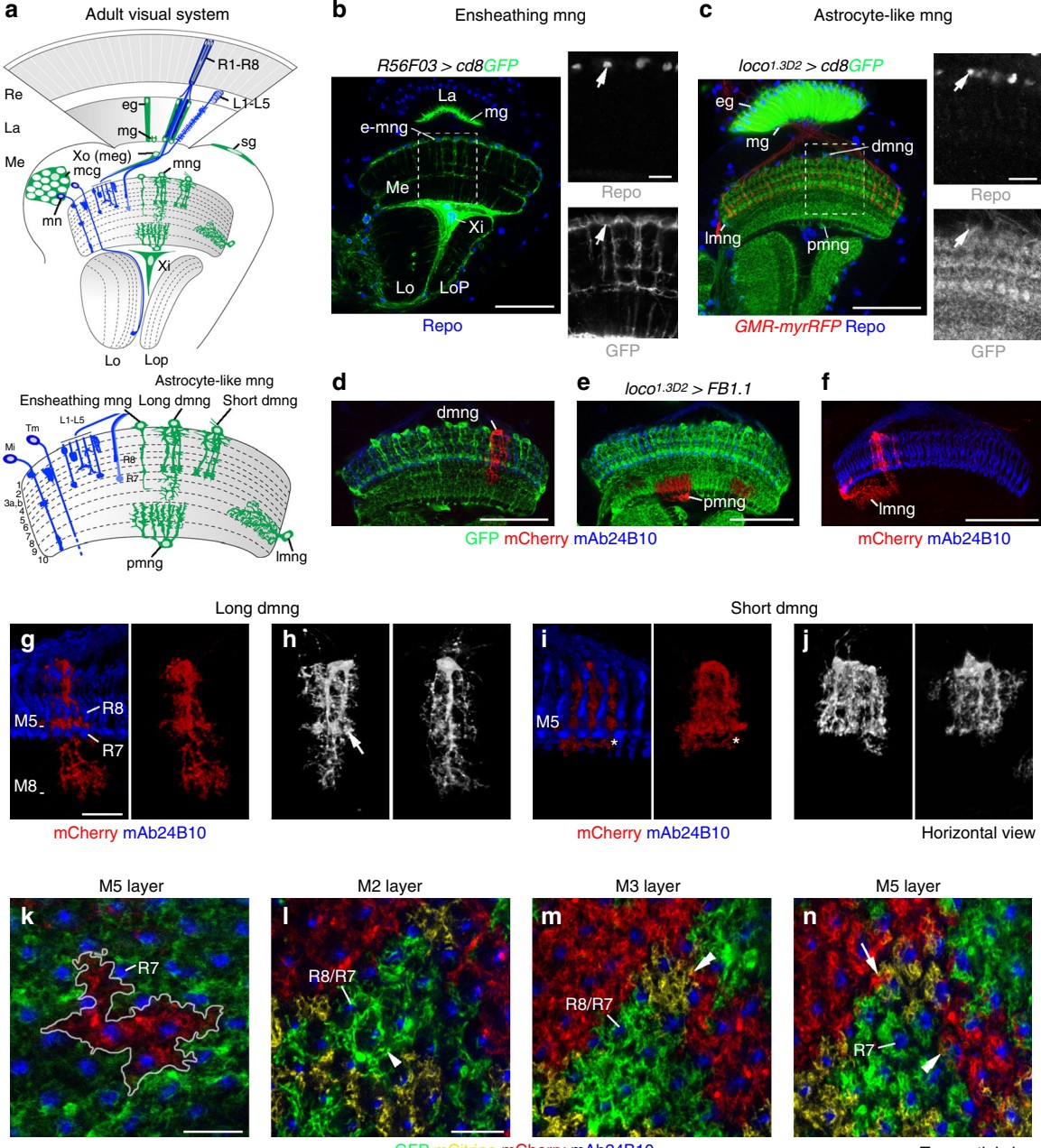

**Fig. 1** Four astrocyte-like glial variants in the medulla. **a** Schematic of glial subtypes in the adult *Drosophila* optic lobe. *dmng* distal medulla neuropil glia, *eg* epithelial glia, *L1-L5* lamina neurons, *La* lamina, *lmng* lateral mng, *Lo* lobula, *Lop* lobula plate, *mcg* medulla cortex glia, *Me* medulla, *mg* marginal glia, *Mi* medulla intrinsic neurons, *mn* medulla neurons, *pmng* proximal mng, *R1-R8, R8/R7* R-cell and axon subtypes, *Re* retina, *sg* surface glia, *Tm* transmedullary neurons, *Xi* inner chiasm glia, *Xo or meg* outer chiasm glia or medulla glia. *R56F03-Gal4* (**b**) and *loco[1.3D2]-Gal4* (**c**) drive expression of *UAS-cd8GFP* (*green*) in ensheathing mng (e-mng) and astrocyte-like mng, respectively. R-cell axons were labeled with *GMR-myrRFP* (**c**, *red*), and glial nuclei with Repo (**b**, **c**, *blue*, *arrows*). **d–n** Single astrocyte-like mng were visualized using *loco[1.3D2]-Gal4* and *UAS-FB1.1*. Processes of dmng infiltrate medulla neuropil layers M1-M8 (**d**), pmng layers M9 and M10 (**e**) and lmng layers M7 and M8 (**f**). dmng visualized with volocity include long (**g**, **h**) and short (**i**, **j**) variants with stereotypic morphologies terminating in layers M8 and M5, respectively. dmng form microdomains in layer M5 (**h**, *arrow*). Asterisks in **i** indicate branches in layer M5 in a deeper focal plane. **k** Processes of single dmng (*outlined*) associate with 7–8 columns in layer M5. **l–n** Glial branches accumulate in the column periphery in layer M2 (**l**, *arrowhead*), evenly spread through columns in M3 (**m**) and aggregate in the column center adjacent to R7 axons in M5 (**n**). mng branches show a tiled organization (**l–n**). Processes of several mng converge into the same column (**n**, *arrow*, similar to microdomains in **h**) and marginally overlap (**m**, **n**, *double arrowheads*). R-cell axons were labeled with mAb24B10 (*blue*). Panels **b–f**, **g–n** show single optical sections, panels **g–j** show projections. For genotypes and sample numbers, see Supplementary Table 1. *Scale bars*, 50 μm (**b–f**), 20 μm (**g–n**), 10 μm (small panels **b**, **c**)

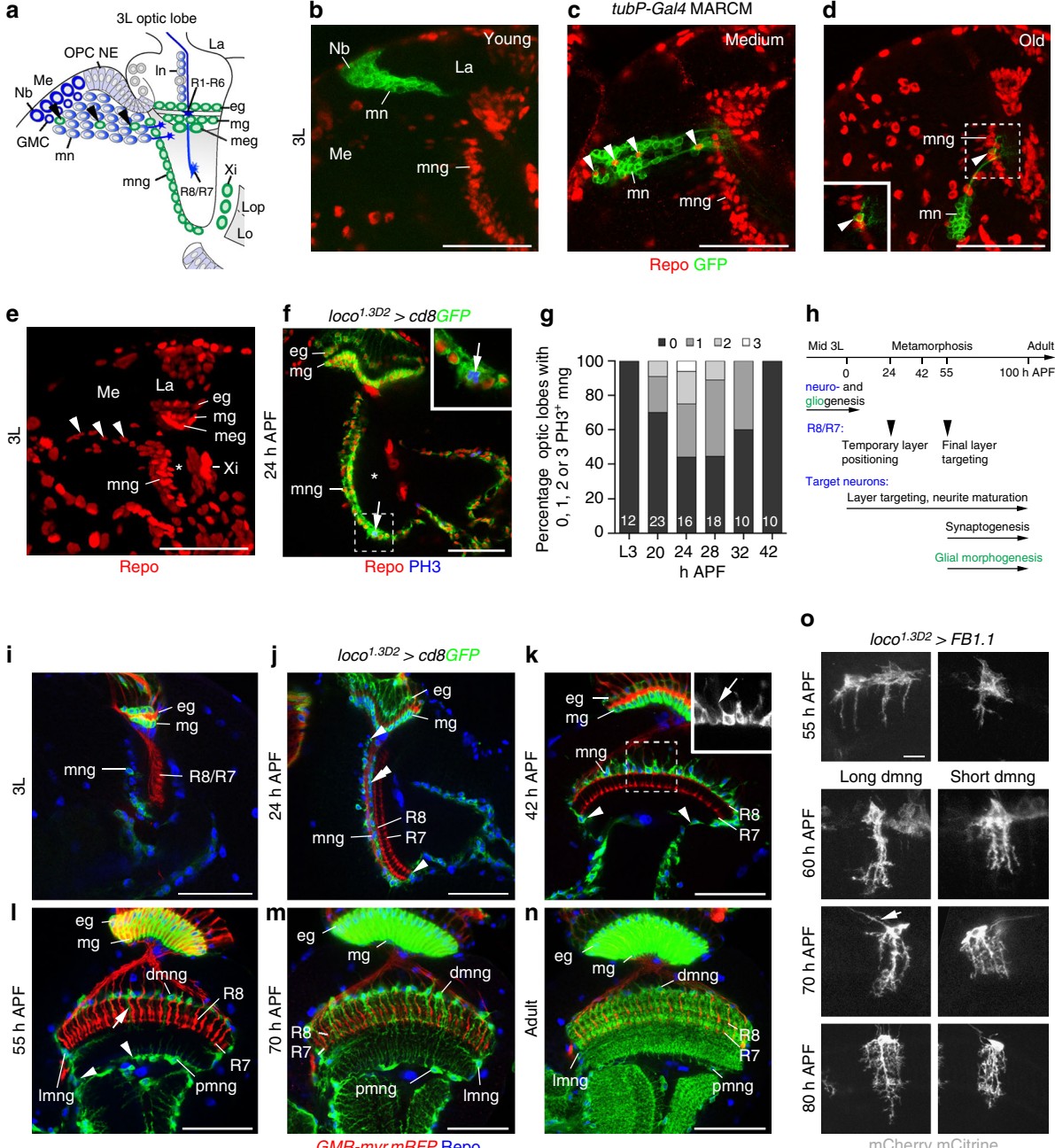

**Fig. 2** Development of astrocyte-like medulla neuropil glia. **a** Schematic of a third instar larval (3L) optic lobe. *mng* medulla neuropil glia, *eg* epithelial glia, *GMC* ganglion mother cell, *La* lamina, *Lo* lobula, *Lop* lobula plate, *ln* lamina neurons, *Me* medulla, *meg* medulla glia, *mg* marginal glia, *mn* medulla neurons, *Nb* neuroblast/neuroglioblast, *OPC NE* outer proliferation center neuroepithelium, *R1-R6, R8/R7*, R-cell axon subtypes, *Xi* inner chiasm glia. *Arrowheads*, migratory mng. **b–d** MARCM clones were generated using *tubP-Gal4* and *hs-FLP* transgenes. Young GFP-labeled clones (*green*) closest to the OPC neuroepithelium contain Nb and neurons (**b**, *n* = 18). Medium (**c**, *n* = 6) and older (**d**, *n* = 11) clones include Repo-positive glial cells (*arrowheads*, *red*). **e** A projection of optical sections illustrates the migratory path of OPC-derived mng (*arrowheads*) through the cortex toward the anterior medulla neuropil edge (*asterisk*). **f** In 24 h APF optic lobes, mng were labeled with *loco^{1.3D2}-Gal4 UAS-cd8GFP* (*green*) and Repo (*red*), and cells undergoing mitosis with phosphoHistone 3 (PH3, *blue*). **g** Percentage of optic lobes containing PH3-positive mng during early metamorphosis (*n* = optic lobes: 12, 23, 16, 18, 10, and 10). **h** Timeline of key steps underlying R-cell, target neuron, and glial development. **i–n** mng were labeled with *loco^{1.3D2}-Gal4 UAS-cd8GFP* (*green*) and Repo (*blue*). During 3L (**i**), at 24 h (**j**) and 42 h APF (**k**), mng cell bodies at the neuropil border (**j**, *double arrowhead*) migrate around the neuropil (**j–l**, *arrowheads*). Distal mng (*dmng*) extend short processes into the cortex at 42 h APF (**k**, *insert*) and into the neuropil from 55 h APF onwards (**l**, *arrows*). These become highly branched at 70 h APF and in adults (**m**, **n**). **o** Long and short dmng variants were labeled with *UAS-FB1.1* (*white*). mng extend filopodia-like processes into the neuropil at 55 h APF. Primary processes project to final layers at 60 h APF. Increasingly branched secondary processes arise at 60, 70, and 80 h APF. Some dmng send thin processes along axon tracts into the cortex (*arrow* at 70 h APF). **Panels f**, **i–n** show single optical sections, **b–e**, **o** projections. For genotypes and sample numbers, see Supplementary Table 1. *Scale bars*, 50 μm (**b–f**, **i–n**), 10 μm (**o**)

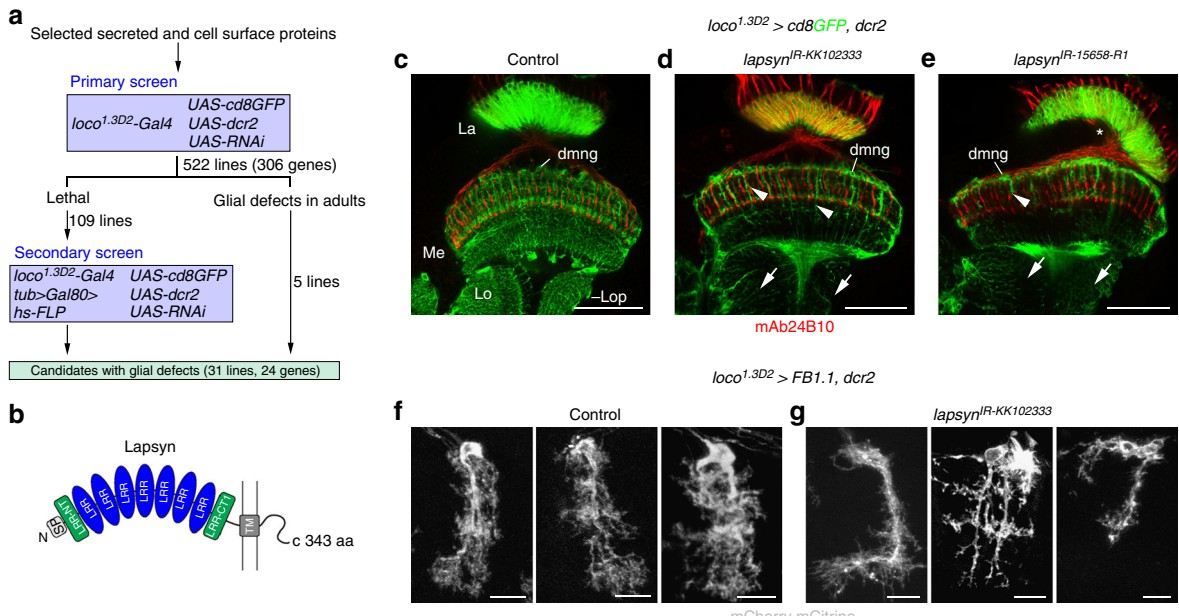

**Fig. 3** A RNA interference screen for factors mediating glial branch morphogenesis. **a** Schematic representation of screen strategy. To knockdown genes that encode selected secreted and cell surface molecules, *UAS-RNAi* and *UAS-dcr2* were expressed in astrocyte-like glia under the control of *loco^1.3D2^-Gal4*. A secondary screen was performed for *UAS-RNAi* causing early lethality. To achieve mosaic expression of *UAS-RNAi* transgenes in astrocyte-like glia using *loco^1.3D2^-Gal4*, a heat shock-FLP (*hs-FLP*) recombinase transgene was induced transiently to promote the random excision of the *FRT* site flanked Gal4-repressor Gal80 encoding cDNA downstream of the widely active tubulin enhancer (*tub > Gal80>*)[68]. **b** *Lapsyn* encodes a transmembrane (TM) cell adhesion protein containing seven leucine-rich repeat (LRR) domains flanked by N-terminal LRR-NT and C-terminal LRR-CT1domains. *SP* signal peptide. **c–e** *loco^1.3D2^-Gal4* drives expression of *UAS-cd8GFP*, *UAS-dcr2*, and two independent RNAi constructs, *UAS-lapsyn^IR-KK102333^* and *UAS-lapsyn^IR-15658-R1^*. **f, g** *loco^1.3D2^-Gal4* drives expression of *UAS-FB1.1* (*white*) in controls (**f**) and *UAS-lapsyn^IR-KK102333^*, *UAS-dcr2* in *lapsyn* knockdown animals (**g**). *La* lamina. Compared to controls (**c, f**), processes of astrocyte-like mng are reduced following *lapsyn* knockdown (**d, e, g**) in the medulla (Me) (**d, e**, *arrowheads*), lobula (Lo), and lobula plate (Lop) (**d, e**, *arrows*). Some brains displayed a medulla rotation defect (**e**, *asterisk*). Panels **c–e** show single optical sections, **f, g** projections. For genotypes and sample numbers, see Supplementary Table 1. Scale bars, 50 μm (**c–e**) and 10 μm (**f, g**)

(PH3) labeling indicated that OPC neuroglioblasts are the primary source for mng because these do not undergo mitosis during the third instar larval stage and only divide occasionally between 20 and 30 h after puparium formation (APF; Fig. 2f, g).

During metamorphosis, R8 and R7 axons initially reside in temporary layers and target to their final layers M3 and M6 around 55 h APF[27]. Target neuron subtypes acquire their characteristic branching patterns throughout pupal development[27, 43]. Generation of mature synapses in the visual system is initiated during mid-pupal development[44] (Fig. 2h). We observed that larval mng begin to express *UAS-cd8GFP* under the control of *loco^1.3D2^-Gal4* after they completed migration to the neuropil border (Fig. 2i). During the first half of metamorphosis, cell bodies spread to the lateral and proximal medulla neuropil border (Fig. 2j–l). Astrocyte-like mng send transient short processes into the cortex at 42 h APF and longer branches into the neuropil from 55 h APF onwards (Fig. 2k–n). Flybow labeling (Fig. 2o) revealed that dmng initially form filopodia-like processes, which transform into main columnar branches characteristic for long and short variants to then generate fine layered secondary branches. Thus, stepwise branch morphogenesis of astrocyte-like mng is spatially and temporally controlled, and takes place after R-cell axons have reached their final layers, in parallel with target neuron arbor maturation and synapse formation.

**A screen for factors regulating mng branch formation.** Glial branch morphogenesis is likely mediated by cell-cell interactions. We therefore sought to identify secreted and cell surface proteins

that control branch formation of astrocyte-like mng in a high-resolution confocal-based RNA interference (RNAi) screen. *loco^1.3D2^-Gal4*, *UAS-dicer2* (*dcr2*) and *UAS-cd8GFP* were crossed with selected *UAS-RNAi* transgenes to achieve efficient knockdown and to visualize astrocyte-like mng in pharate adults. Screening 522 *UAS-RNAi* lines directed against the mRNA of 306 genes, we uncovered 24 genes whose RNAi-mediated knockdown caused defects (Fig. 3a).

Candidates with strong branching defects included a member of the extracellular leucine-rich repeat (LRR) protein superfamily encoded by *lapsyn* (leucine-rich repeat activity-regulated protein at synapses)[33]. The 343 amino acid (aa) protein consists of an extracellular domain with seven central LRR motifs flanked by N-terminal LRR-NT and C-terminal LRR-CT1 motifs, followed by a transmembrane region and a short 45 aa intracellular sequence of low complexity that lacked homology with known domains[45] (Fig. 3b). *lapsyn* has been identified as a gene whose transcript levels were altered in seizure- and learning and memory-mutants[33]. We found that *lapsyn* knockdown in astrocyte-like mng severely impaired the infiltration of the adult medulla neuropil by glial processes of dmng, pmng, and lmng (Fig. 3c–e). Phenotypes were highly penetrant for two independent *UAS-RNAi* transgenes (*UAS-lapsyn^IR-KK102333^*: 74%, $n = 19$; *UAS-lapsyn^IR-15658-R1^*: 62%, $n = 13$). dmng, individually labeled with Flybow ($n = 61$ in 22 samples), showed variable phenotypes (Fig. 3f, g): this included abnormally thick or fasciculated primary and secondary branches (24.5%; Fig. 3g, *left panel*), main processes with reduced secondary branches (37.8%; Fig. 3g, *middle panel*) or short unbranched primary processes (26.2%; Fig. 3g, *right panel*).

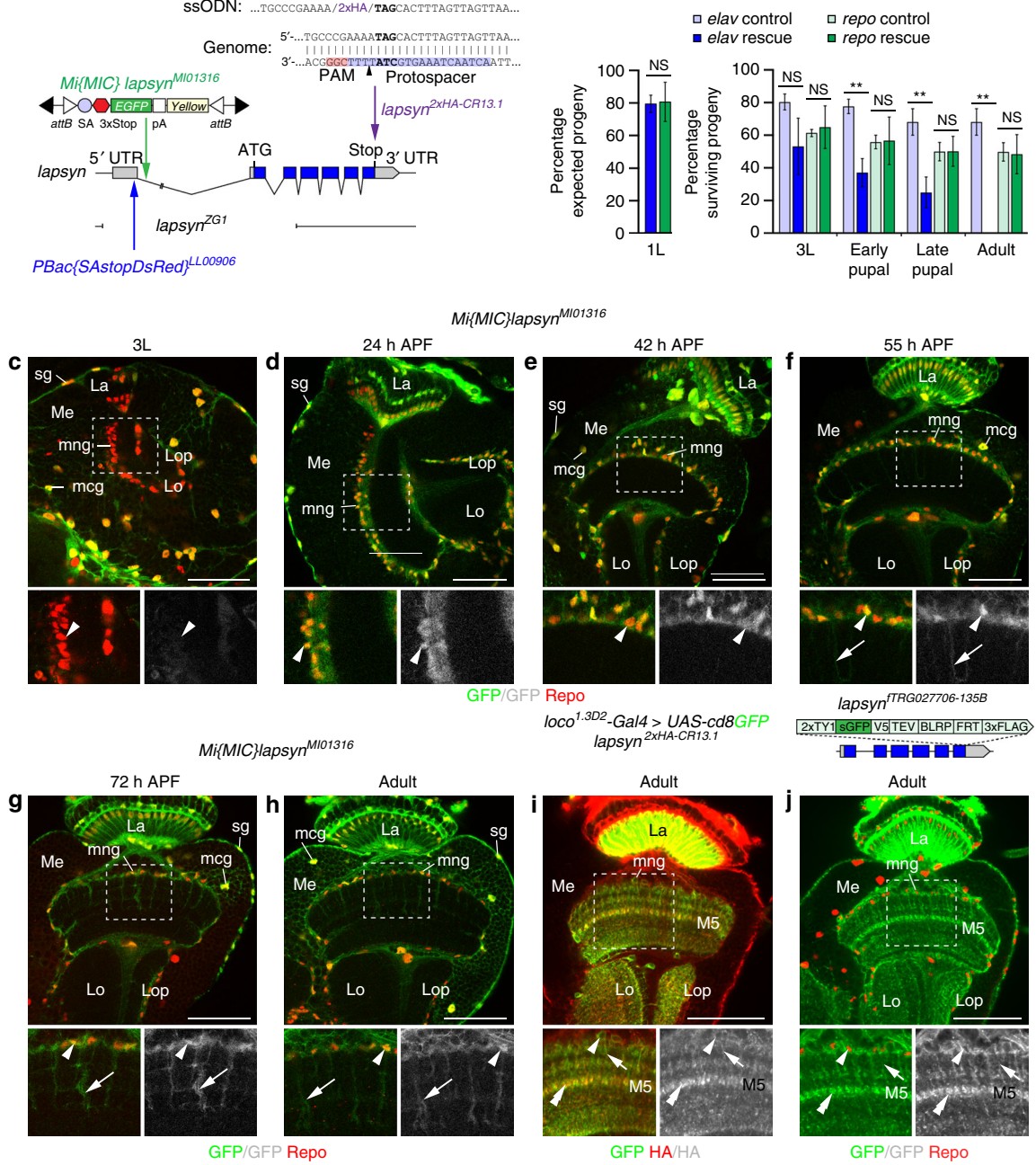

**Fig. 4** *lapsyn* is expressed in glia. **a** The schematic shows the *lapsyn* genomic locus, the *lapsyn^{ZG1}* deletion, and *Mi{MIC}lapsyn^{MI01316}* and *PBac{SAstopDsRed}^{LL00906}* insertions. *pA* polyadenylation site, *SA* splicing acceptor site. To generate *lapsyn^{2xHA-CR13.1}*, two C-terminal HA epitope tags were inserted into the endogenous locus following a Cas9-induced DSB and HDR with a single-stranded oligo donor nucleotide (*ssODN*) containing flanking 50-nt homology arms. **b** For whole animal rescue experiments of *lapsyn^{ZG1}* embryonic lethality, *UAS-lapsyn* was expressed in neurons (*elav^{C155}-Gal4*, *blue bars*, 224 recovered first instar (1L) larvae of 1210 embryos) or in glia (*repo-Gal4*, *green bars*, 529 recovered 1L larvae of 2380 embryos). GFP-negative rescue flies were compared to controls carrying *CyO Dfd-YFP* balancer chromosomes, corresponding to all other surviving progeny (heterozygous, or heterozygous and overexpressing *lapsyn*, *light blue* and *green bars*). Rescue of viability of 1L larvae (*left panel*) is shown as percentage of expected progeny. Rescue of viability of third instar (3L) larvae, early and late pupae and adults is shown as percentage of surviving progeny in "rescue" and "control" populations followed from the 1L stage onwards. *Histograms* show data points as means ± standard deviation error bars (*n* = 3 independent experiments). Unpaired, two-tailed Student's *t*-test assuming normality but unequal standard deviation (1L: *P* = 0.879; 3L: *P* = 0.1027, *P* = 0.6796; early pupae: *P* = 0.006, *P* = 0.9369; late pupae: *P* = 0.0042, *P* = 0.9921; adult: *P* = 0.0047, *P* = 0.8668). *NS* not significant, \*\**P* < 0.01. **c–h** *Mi{MIC}lapsyn^{MI01316}* is inserted into the first non-coding intron. In 3L larvae, Repo-positive (*red*) surface glia (*sg*) and medulla cortex glia (mcg), but not mng (**c**) are labeled with GFP. mng express GFP at 24, 42, 55, and 72 h APF, and in adults (*arrowheads*). mng processes are labeled from 55 h APF onwards (**f–h**, *arrows*). **i, j** *lapsyn^{2xHA-CR13.1}* and the fosmid *lapsyn^{fTRG027706}* report abundant lapsyn protein localization in mng cell bodies (*arrowheads*), primary (*arrows*), and secondary processes (*double arrowheads*), particularly in layer M5, using anti-HA (*red*) and GFP labeling (*green*), respectively. *La* lamina, *Lo* lobula, *Lop* lobula plate, *Me* medulla. Panels **c–j** show single optical sections. For genotypes, sample numbers and additional statistical values, see Supplementary Tables 1 and 2. *Scale bars*, 50 μm

Next, we examined the cytoskeleton of adult astrocyte-like mng. Using $loco^{1.3D2}$-Gal4, we co-expressed membrane-bound GFP or mCherry with fluorescent protein-tagged cytoskeletal markers (Supplementary Fig. 3). In control astrocyte-like mng, the microtubule (MT) marker mCherry-α-Tubulin (ChRFP-Tub) and the MT plus-end tracking fusion protein Eb1-GFP were mainly detected in primary branches, whereas nod-GFP, a minus-end MT marker, accumulated in cell bodies. LifeAct-GFP revealed actin-rich secondary branches. Thus, similar to VNC astrocyte-like glia[26], mng showed a differential distribution of MT and actin in their processes. Following lapsyn knockdown, actin expression was severely reduced, whereas MT appeared less affected, consistent with the loss of mainly secondary branches.

**Lapsyn is expressed in astrocyte-like mng.** To distinguish whether lapsyn acts in glia and neurons or solely in glia, we conducted cell-type specific rescue experiments monitoring the viability of progeny during development. $lapsyn^{ZG1}$, a null allele carrying a ~4.7 kb deletion, is homozygous embryonic lethal[33] (Fig. 4a). Overexpression of lapsyn with the pan-neuronal driver $elav^{C155}$-Gal4 partially rescued the lethality of $lapsyn^{ZG1}$ mutants up to the third instar larval stage, but these larvae barely moved (Fig. 4b). Their viability strongly decreased in pupae, and no adults emerged. In line with previous findings[33], suggesting a non-neuronal requirement of lapsyn, expression with the pan-glial driver repo-Gal4 substantially restored viability of progeny throughout development and resulting adults could be maintained as a fertile stock.

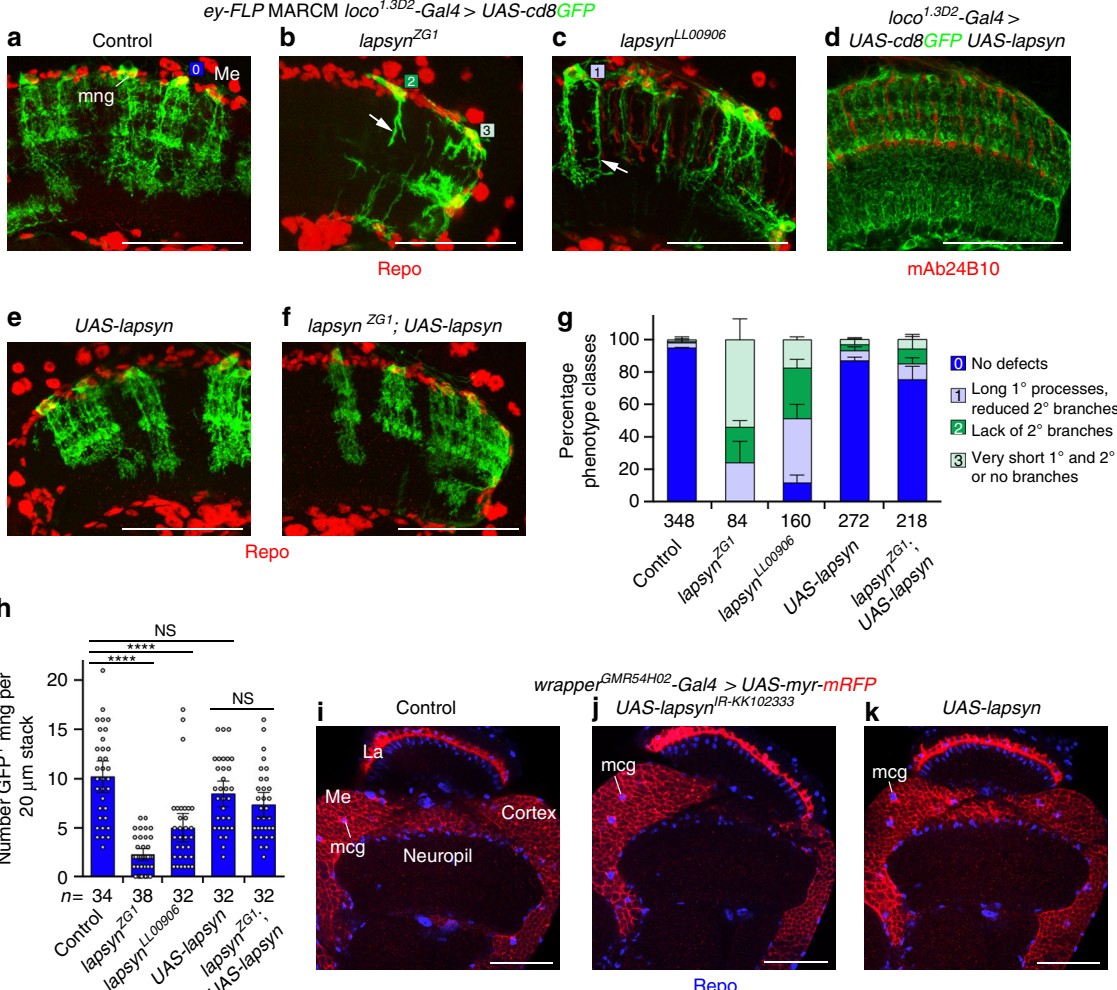

**Fig. 5** lapsyn is cell-autonomously required for branch morphogenesis of astrocyte-like mng. **a–c** Control, $lapsyn^{ZG1}$ and $lapsyn^{LL00906}$ mutant astrocyte-like mng clones were generated by combining ey-FLP and MARCM. Clones were labeled with $loco^{1.3D2}$-Gal4 UAS-cd8GFP (green) and all glial nuclei with Repo (red). Unlike control clones displaying a highly elaborate morphology (**a**, square-0), lapsyn mutant astrocyte-like mng showed severe defects: primary and secondary processes were very short (**b**, square-1), secondary branches were missing (**b**, square-2) or reduced (**c**, square-3). **d, e** Overexpression of UAS-lapsyn with $loco^{1.3D2}$-Gal4 in all astrocyte-like glia (**d**) or in MARCM clones (**e**) did not affect general mng morphology. **f** Overexpression of UAS-lapsyn rescued morphological defects in lapsyn mng clones. **g** Quantification of phenotype classes. 100% stacked bar histograms show data points as means plus standard deviation error bars (n = independent experiments assessing individual mng in 20-μm optical stacks/optic lobes: 3 [34/19], 3 [38/24], 2 [32/18], 3 [32/17], 3 [31/16]). Color code same as in **a–c**. **h** Quantification of GFP-positive astrocyte-like mng numbers. In adults, fewer clones were recovered in lapsyn mosaics compared to controls. Overexpression of UAS-lapsyn does not affect cell numbers and rescues the reduction of clones in $lapsyn^{ZG1}$. The scatter plot with bars shows data points and means ± 95% confidence intervals (n = 20-μm stacks of optical sections from 19,24,18,17, and 16 optic lobes). Unpaired, two-tailed Student's t-test not assuming equal variance: $P = 2.93 \times 10^{-12}$, $P = 6.44 \times 10^{-6}$, $P = 0.0882$, $P = 0.2131$. NS not significant, ****$P < 0.0001$. **i–k** Knockdown and overexpression of lapsyn in medulla cortex glia (mcg) using $wrapper^{GMR54H02}$-Gal4 to express UAS-lapsyn[IR-KK102333] (**j**) or UAS-lapsyn (**k**) does not affect their morphology compared to controls (**i**). La lamina, Me medulla. Panels **d, i–k** show single optical sections, **a–c, e, f** projections. For genotypes, sample numbers and additional statistical values, see Supplementary Tables 1 and 2. Scale bars, 50 μm (**a–f, i–k**)

Next, we examined the expression of Lapsyn. Because previously generated[33] and our own antisera were not suitable for tissue immunolabeling, we tested a Minos-mediated integration cassette (MiMIC) insertion[46] into *lapsyn*. *Mi{MIC}lapsyn[MI01316]* is inserted in a 5′ to 3′ orientation within the first intron between 5′UTR sequences, thus reporting GFP expression in the cells producing Lapsyn (Fig. 4a). We detected expression in

glia but not neurons. Repo-positive surface and cortex glia were labeled throughout development, whereas neuropil-associated glial cell types, including mng, showed expression from 24 h APF onwards (Fig. 4c–h).

To determine the localization of Lapsyn protein, we initially used a φC31 recombinase-mediated cassette exchange (RMCE) strategy[46], replacing *Mi{MIC}lapsyn[MI01316]* with *lapsyn* cDNA and two

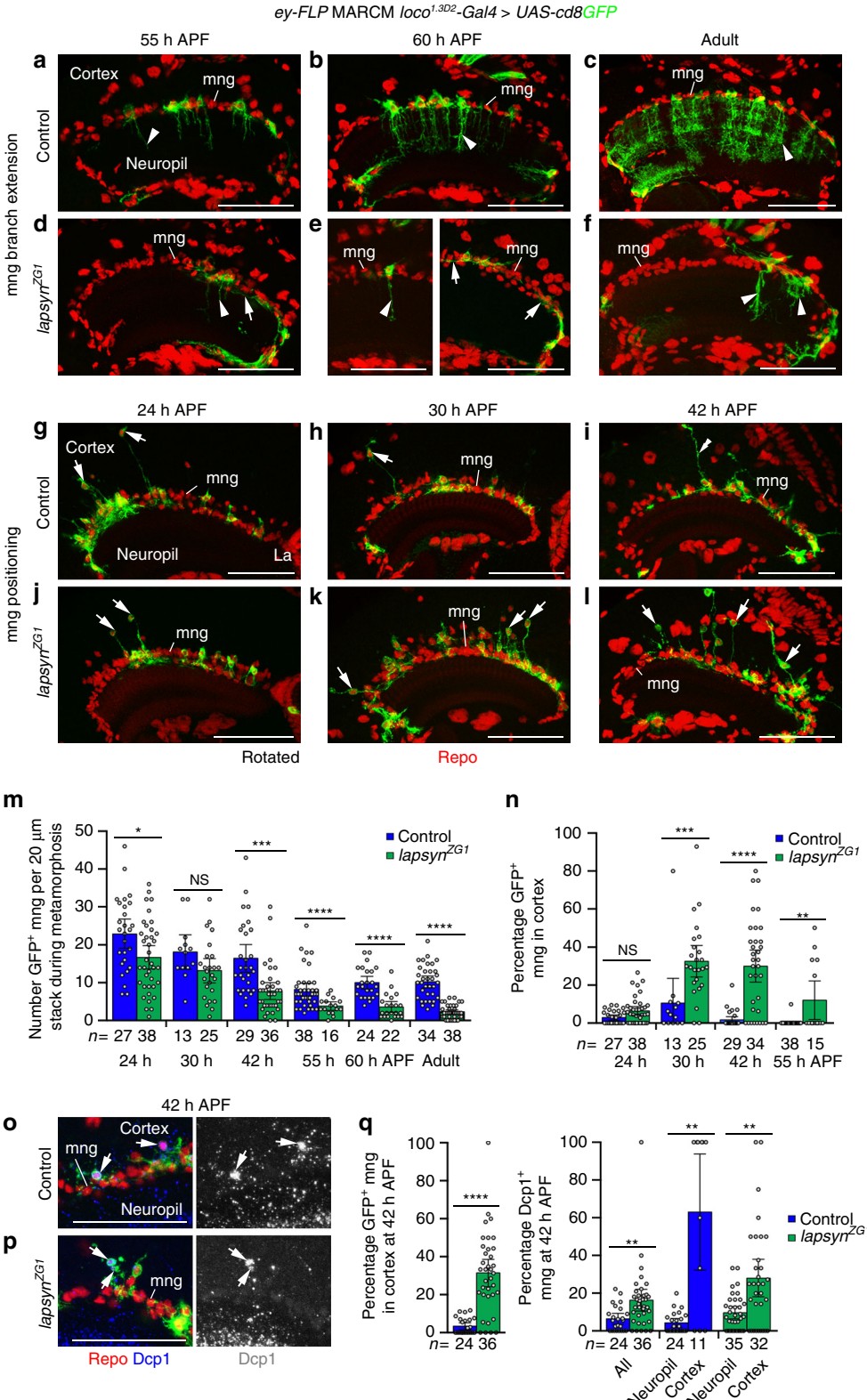

C-terminal hemagglutinin (HA) epitope-tag encoding sequences. While the resulting insertion $Mi\{MIC\}lapsyn^{lapsyn-2xHA-24A}$ reported protein expression in astrocyte-like mng (Supplementary Fig. 4a–c), levels were low. We therefore pursued a Crispr/Cas9-mediated protein-tagging approach[47]. We inserted two in-frame C-terminal HA-encoding sequences into the *lapsyn* locus by inducing a Cas9-mediated double-stranded break (DSB) upstream of the stop codon while promoting homology directed repair (HDR) with a single-stranded oligo donor nucleotide. The template consisted of two 50-nt homology arms flanking the HA sequences (Fig. 4a). In parallel, we generated a transgenic line carrying a C-terminally tagged *lapsyn* fosmid from the FlyFos library[48] (Fig. 4j). In the resulting $lapsyn^{2xHA-CR13.1}$ Crispr-Cas9 and $lapsyn^{fTRG027706}$ fosmid lines, endogenous Lapsyn tagged with HA and GFP, respectively, was clearly detectable in the processes of adult astrocyte-like mng (Fig. 4i, j). Finally, we observed punctate Lapsyn in adult mng processes when overexpressing a *UAS-lapsyn-GFP* transgene[33] (Supplementary Fig. 4d). Together, these approaches consistently report that Lapsyn is glial-specific in the optic lobe and located in the processes of astrocyte-like mng.

### *lapsyn* cell-autonomously controls mng branch formation.

To test whether or not *lapsyn* is cell-autonomously required in astrocyte-like mng, we generated MARCM clones with two homozygous embryonic lethal alleles, $lapsyn^{ZG1}$ and $lapsyn^{LL00906}$, a 5′UTR *piggyBAC* insertion[49] (Fig. 4a). *eyeless-FLP* (*ey-FLP*) served as the recombinase source in OPC neuroblasts to remove *lapsyn* in newborn mng. $loco^{1.3D2}$-*Gal4* was used to unambiguously identify astrocyte-like mng. In line with knockdown experiments, adult *lapsyn*-deficient mng displayed three classes of defects: failure to extend processes into the neuropil, formation of short unbranched primary processes and extension of long primary processes with reduced secondary branches (Fig. 5a–c, g).

Because clones included glia and neurons, we next conducted rescue experiments. Expression of untagged Lapsyn[33] in all astrocyte-like mng or in MARCM clones did not affect dmng branch formation or territory size (Fig. 5d, e, g). Expression of Lapsyn in $lapsyn^{ZG1}$ mutant mng substantially rescued the morphological defects caused by *lapsyn* loss (Fig. 5f, g). The number of clones at the medulla neuropil border was lower for $lapsyn^{ZG1}$ and $lapsyn^{LL00906}$ mutants than controls. Overexpression of *lapsyn* rescued also this phenotype (Fig. 5h). Hence, *lapsyn* is cell-autonomously required in astrocyte-like mng.

Monitoring the behavior of cells adjacent to *lapsyn*-deficient astrocyte-like mng, we detected processes of heterozygous mng in close vicinity of the sparse branches of mutant mng (Supplementary Fig. 5; 69/69 cells in 20 samples). This suggests that

astrocyte-like mng maintain their numbers and/or adjust branch growth for complete neuropil coverage. To assess a potential role in other glia, we manipulated *lapsyn* in medulla cortex glia using $wrapper^{GMR54H02}$-*Gal4*[35]. Knockdown did not reduce branch formation, and overexpression did not enhance branch extension within the cortex or direct processes into the medulla neuropil (Fig. 5i–k). Thus, *lapsyn* is neither required in cortex glia nor sufficient to control branch morphogenesis, suggesting that co-factors, found in mng, may be absent in this glial subtype.

To determine the developmental time window during which *lapsyn* is required in astrocyte-like mng for branching, we compared wild type and mutant clones during metamorphosis (Fig. 6a–f). At 55 h APF, the first wild-type mng began to extend columnar processes into the medulla neuropil. By contrast, *lapsyn*-deficient mng either extended very short branches into the neuropil or formed branches that projected between glial cell bodies across the neuropil surface (Fig. 6a, d). Defects persisted at 60 h APF, when all control mng have sent branches into the neuropil (Fig. 6b, e) and in adults (Fig. 6c, f). Hence, *lapsyn* controls branch formation during mid-pupal development rather than maintenance at later stages.

### Lapsyn controls mng positioning at the neuropil border.

To shed light on the potential causes for the reduction in recovered adult *lapsyn* mutant clones (Fig. 5h), we next examined the distribution and numbers of astrocyte-like mng during metamorphosis. Unexpectedly, in both control and $lapsyn^{ZG1}$ mosaics, we observed GFP-positive mng cell bodies in the medulla cortex at 24, 30, and 42 h APF (Fig. 6g–l), but not in adults (Fig. 6c, f). At 24 h APF, mng numbers at the distal medulla neuropil border were only slightly lower in $lapsyn^{ZG1}$ mosaics than in controls, suggesting that *lapsyn* is not essential for their differentiation or initial migration to the neuropil border (Fig. 6m). In control and experimental mosaics, the number of recovered clones decreased until ~55 h APF and then remained constant. This decrease was more pronounced in $lapsyn^{ZG1}$ mosaics. In the cortex at 24 h APF, the percentages of control and *lapsyn*-deficient mng were low and similar. However, at 30, 42, and 55 h APF, they were considerably higher for mutant mng than controls (Fig. 6n). Cortical localization correlated with a higher percentage of cells labeled with an antibody against cleaved *Drosophila* death caspase-1 (Dcp1) (Fig. 6o–q). While overexpression of the anti-apoptotic Baculo virus gene *p35* in $lapsyn^{ZG1}$ clones did not rescue mng numbers at the neuropil border (Supplementary Fig. 6a, b), we detected 8% surviving mng in the adult cortex compared to 0% in controls ($lapsyn^{ZG1}$, *UAS-p35*: 100 mng in 13 samples; *UAS-p35*: 293 mng in 12 samples). These findings suggest that after their initial migration to the medulla neuropil border, mng retain motile properties and require *lapsyn* for

**Fig. 6** *lapsyn* controls astrocyte-like mng branch formation and positioning during development. Clones were generated using *ey-FLP* MARCM and visualized with $loco^{1.3D2}$-*Gal4 UAS-cd8GFP* (*green*). Repo-labeled glial nuclei (*red*). **a–f** Assessment of branch formation. Unlike in controls, $lapsyn^{ZG1}$ mng extended reduced branches into the neuropil (*arrowheads*) or ectopic branches between glial cell bodies at the neuropil border (*arrows*) at 55 and 60 h APF. Defects persisted in adults. **g–l** Assessment of positioning. Compared to controls, more $lapsyn^{ZG1}$ mng cell bodies were detected in the cortex at 30 and 42 h than at 24 h APF (*arrows*). These sent long protrusions to the neuropil border. mng at the border extended long processes into the cortex (**i**, *double arrowhead*). Quantifications assessing numbers of GFP-positive mng (**m**) and percentages in the cortex of control and $lapsyn^{ZG1}$ mosaics (**n**) during metamorphosis. Scatter plots with bars show data points and means ± 95% confidence intervals (*n* = 20-μm stacks of optical sections from 19,27,6,12,18,21,22,12,13,13,19, and 23 (**m**) and 19, 27, 6, 12, 18, 19, 22, and 12 optic lobes (**n**). **m** Unpaired, two-tailed Student's *t*-test not assuming equal variance: $P = 0.0129$, $P = 0.0678$, $P = 0.0001$, $P = 5.47 \times 10^{-5}$, $P = 4.38 \times 10^{-7}$, $P = 2.03 \times 10^{-12}$. **n** Two-tailed Mann-Whitney *U*-test corrected for ties: $P = 0.0557$, $P = 0.0002$, $P = 4.0 \times 10^{-7}$, $P = 0.0010$. **o, p** Dcp1 (*blue*) labeling reveals that control (**o**) and $lapsyn^{ZG1}$ (**p**) mng in the cortex undergo apoptosis (*arrows*) at 42 h APF. **q** Comparison of percentages of control and $lapsyn^{ZG1}$ clones in the cortex (*left*) and Dcp1-positive clones in the cortex and at the neuropil border (*right*) at 42 h APF. Scatter plots with bars show data points and means ± 95% confidence intervals (*n* = 20-μm stacks from 21, 29 (*left*), 21, 29, 21, 11, 28, and 26 optic lobes). Two-tailed Mann-Whitney *U*-test corrected for ties: $P = 2.4 \times 10^{-7}$, $P = 0.0016$, $P = 0.0029$, $P = 0.0025$. NS not significant, *$P < 0.05$, **$P < 0.01$, ***$P < 0.001$, ****$P < 0.0001$. Panels **a–l** show projections. For genotypes, sample numbers and additional statistical values, see Supplementary Tables 1 and 2. *Scale bars*, 50 μm

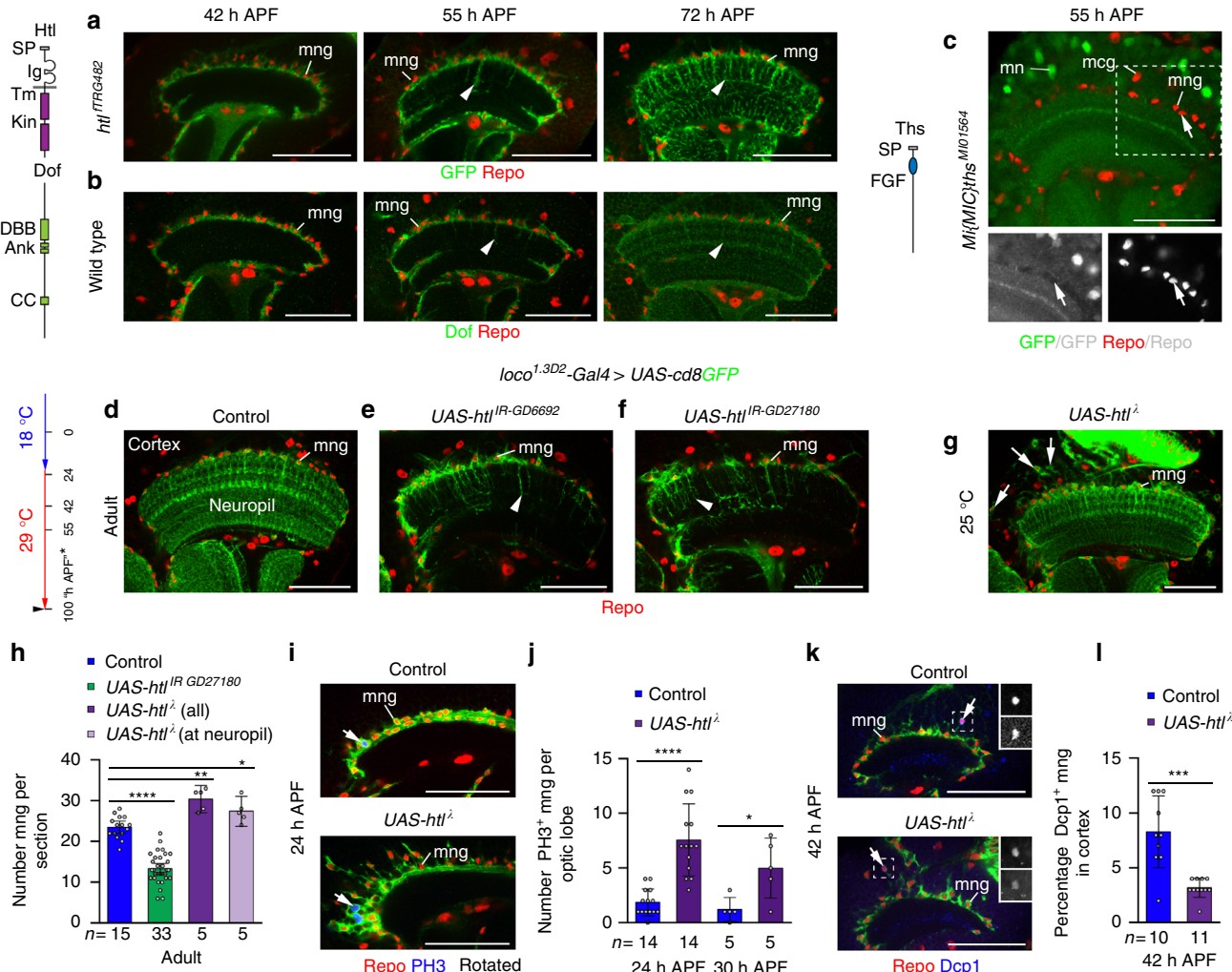

**Fig. 7** FGF signaling regulates astrocyte-like mng development. Repo labeled glial nuclei (*red*). **a** *htl*[fTRG482] reports Htl expression (*green*) at 42, 55, and 72 h APF in mng cell bodies and processes (*arrowheads*). **b** Dof (*green*) labeling of wild-type[OreR] optic lobes showed a similar expression pattern. **c** At 55 h APF, *Mi{MIC}ths*[MI01564] reports Ths production in medulla neurons (MN), but not glia (*arrows*) including mng or medulla cortex glia (MCG). *Ank* ankyrin repeat, *DBB* Dof-BVAP-BANK domain *CC* coiled-coil domain, *Ig* immunoglobulin domain, *Kin* kinase, *SP* signal peptide, *Tm* transmembrane domain. **d–g, l, k** mng were labeled with *loco*[1.3D2]-Gal4 UAS-cd8GFP (*green*). Following expression of *UAS-htl*[IR-GD6692] (**e**) and *UAS-htl*[IR-GD27180] (**f**) adult mng showed reduced primary and secondary processes in the neuropil (*arrowheads*) compared to controls (**d**). Samples were shifted to 29 °C at 21 h APF. *Asterisk*, time points are adjusted to h APF at 25 °C. **g** Expression of constitutively active *htl*[λ] in mng did not affect branch extension into the neuropil. Surviving mng are ectopically positioned in the medulla cortex (**g**, *arrows*). **h** Quantification of mng numbers. Htl knockdown decreases and over-activation increases mng numbers. The scatter plot with bars shows data points and means ± 95% confidence intervals (*n* = optic lobes (single optical sections)). **i, j** Numbers of mitotic PH3-positive mng (*blue*) increased following Htl over-activation at 24 and 30 h APF. **k, l** The percentage of apoptotic Dcp1-positive mng (*blue, bottom insets*) per optic lobe in the cortex decreased following Htl over-activation at 42 h APF. The scatter plots with bars **j, l** show data points and means ± standard deviation (*n* = entire optic lobes). Unpaired, two-tailed Student's *t*-test not assuming equal variance were: $P = 9.84 \times 10^{-13}$, $P = 0.0017$, $P = 0.0372$ (**h**), $P = 1.39 \times 10^{-5}$, $P = 0.0327$ (**j**), and $P = 0.0007$ (**l**). *$P < 0.05$, **$P < 0.01$, ***$P < 0.001$, ****$P < 0.0001$. All image panels show single optical sections. For genotypes, sample numbers and additional statistical values, see Supplementary Tables 1 and 2. Scale bars, 50 μm

anchoring their cell bodies in this location to ensure their survival.

**FGF signaling but not *lapsyn* controls mng survival.** Does *lapsyn* mediate mng survival directly or indirectly by ensuring exposure to a gliotrophic signal through correct positioning? To address this question, we focused on FGF signaling because of its central conserved role in controlling glial development[23–26] and because we had identified the Htl receptor as one candidate in our screen. In *Drosophila*, binding of the FGF8-like ligands Pyramus and Thisbe (Ths) activates Htl followed by tyrosine phosphorylation of the receptor and the constitutively bound adaptor

protein Downstream of FGF (Dof)[50]. Since the function of FGF signaling in optic lobe neuropils was not known, we first examined the expression of pathway components in the medulla. The *htl*[fTRG482] fosmid transgenic line[48] and an antibody against Dof[25] reported strong expression of both in mng cell bodies throughout development and in processes extending into the neuropil from 55 h APF onwards (Fig. 7a, b). By contrast, expression analysis of *Mi{MiC}ths*[MI01564], a 5′-3′ insertion into the first intron between 5′ UTR sequences, indicates that the *ths* ligand is generated by neurons (Fig. 7c).We then assessed the requirement of *htl* using RNAi-mediated knockdown in astrocyte-like mng. Expression of *UAS-htl*[IR-GD6692] or *UAS-htl*[IR-GD27180] using *loco*[1.3D2]-Gal4 at 29 °C from 24 h after egg laying severely affected mng survival

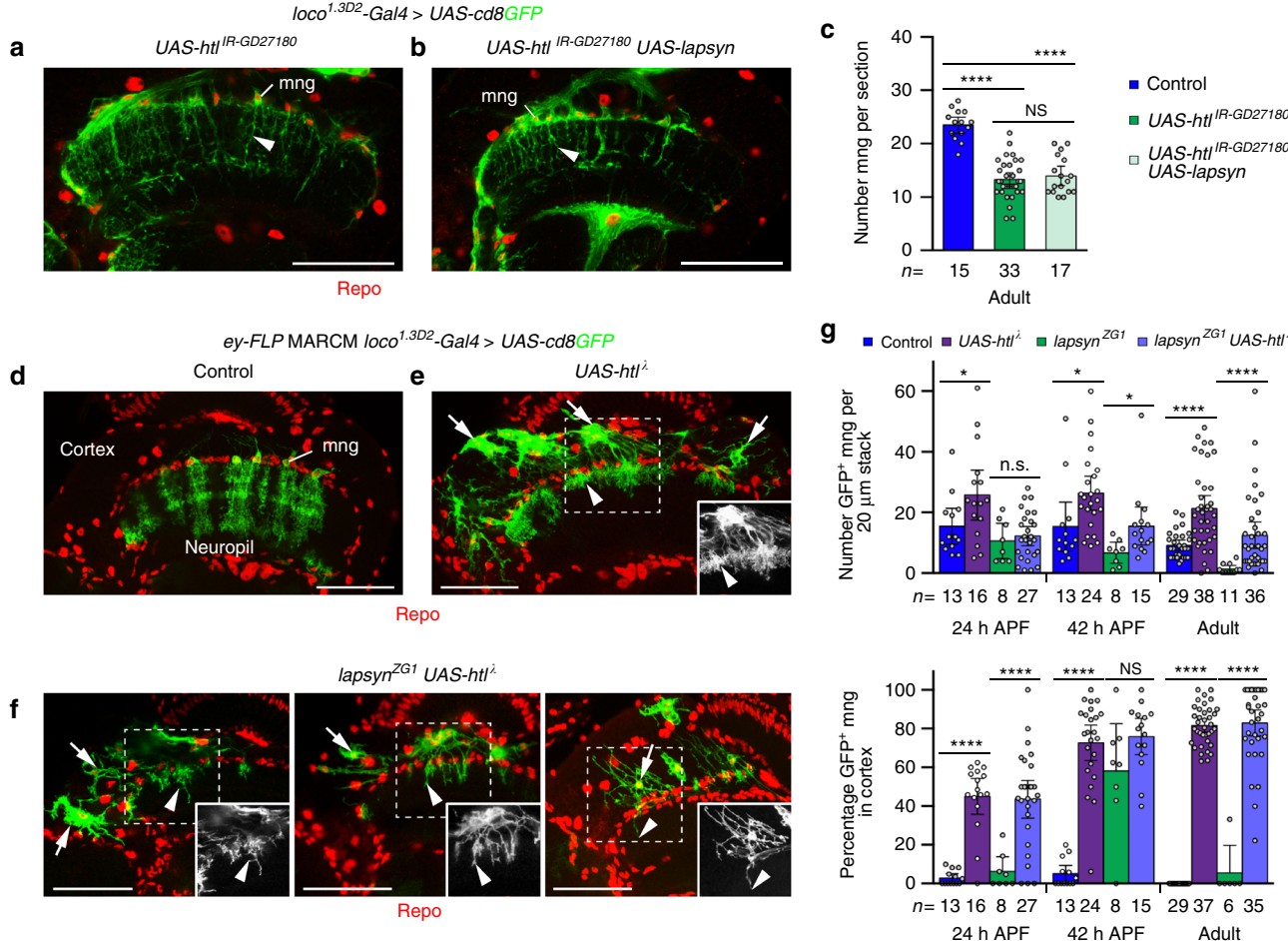

**Fig. 8** *lapsyn* does not directly mediate survival of astrocyte-like mng. Repo-stained glial nuclei (*red*). **a**, **b** mng were labeled with *loco^{1.3D2}*-Gal4 UAS-cd8GFP (*green*). *htl* knockdown with *UAS-htl^{IR-GD27180}* disrupted mng branch morphogenesis in adults (**a**, *arrowhead*). *UAS-lapsyn* expression did not rescue these defects (**b**, *arrowhead*). **c** Quantification of mng numbers. *UAS-lapsyn* expression did not rescue the decrease of cell numbers following *htl* knockdown. The scatter plot with bars shows data points and means ± 95% confidence intervals (n = optic lobes (one optical section)). Unpaired, two-tailed Student's *t*-test not assuming equal variance: $P = 9.84 \times 10^{-13}$, $P = 1.82 \times 10^{-9}$, $P = 0.5166$. **d–f** *UAS-htl^{λ}* was expressed in mng clones using *ey-FLP* MARCM and *loco^{1.3D2}*-Gal4 UAS-cd8GFP. Unlike in controls (**d**), wild-type clones (**e**), and *lapsyn^{ZG1}* mutant clones (**f**) expressing *UAS-htl^{λ}* migrated into the medulla cortex and survived until adulthood (*arrows*). *UAS-htl^{λ}* expressing clones extended short bushy processes into the neuropil (**e**, *arrowhead*). In *lapsyn^{ZG1}* mng clones expressing *UAS-htl* (**f**), branching is reduced in the cortex (*arrow*) and neuropil (*arrowheads*). **g** Quantification of *ey-FLP* MARCM clone numbers and localization at 24 and 42 h APF, and in adults. *UAS-htl^{λ}* expression increased the number of glial cell clones compared to wild type and promoted *lapsyn^{ZG1}* clone survival. High percentages of wild type and *lapsyn^{ZG1}* clones overexpressing *UAS-htl^{λ}* were found in the medulla cortex. Scatter plots with bars show data points and means ± 95% confidence intervals. Unpaired, two-tailed Student's *t*-test not assuming equal variance: $P = 0.0396$, $P = 0.5881$, $P = 0.0241$, $P = 0.0144$, $P = 2.49 \times 10^{-6}$, $P = 1.04 \times 10^{-5}$ (*top*: n = 20-μm stacks of optical sections from 4,7,2,9,6,10,3,6,14,16,4 and 14 optic lobes); $P = 4.25 \times 10^{-8}$, $P = 2.92 \times 10^{-7}$, $P = 7.97 \times 10^{-15}$, $P = 0.1432$, $P = 2.11 \times 10^{-34}$, $P = 6.59 \times 10^{-22}$ (*bottom*: n = 20-μm stacks from 4,7,2,9,6,10,3,6,14,15,4 and 14 optic lobes). NS, not significant, *$P < 0.05$, ****$P < 0.0001$. All image panels show projections. For genotypes, sample numbers and additional statistical values, see Supplementary Tables 1 and 2. *Scale bars*, 50 μm

(Supplementary Fig. 6c–f). To delay the onset of knockdown, progeny were initially raised at 18 °C and switched to 29 °C at a stage equivalent to 21 h APF at 25 °C. While this allowed many mng to survive to adulthood, they displayed severely reduced primary and secondary branches similar to phenotypes observed following *lapsyn* knockdown or loss (Fig. 7d–f, h). In a converse experiment, we increased FGF signaling in all astrocyte-like mng by overexpressing a constitutively activated form of *htl* (*UAS-htl^{λ}*)[51]. Remarkably, many mng relocated into the cortex under these conditions, consistent with a pro-migratory role of FGF signaling[50]. The total number of mng and the number of mng at the neuropil border were increased in adults (Fig. 7g, h). Quantifications of optic lobes at 24 and 30 h APF labeled with PH3 and at 42 h APF with Dcp1 showed that higher mng numbers were due to increased mitosis and reduced apoptosis

levels (Fig. 7i–l). Hence, Htl plays an additional role in the medulla by promoting glial survival.

We next sought to understand the involvement of *lapsyn* in regulating survival in the context of FGF pathway roles. We observed that phenotypes in astrocyte-like mng expressing solely *UAS-htl^{IR-GD27180}* or both *UAS-lapsyn* and *UAS-htl^{IR-GD27180}* were indistinguishable (Fig. 8a–c). This suggests that *lapsyn* expression alone cannot rescue cell loss or branching defects caused by reduced FGF signaling. When overexpressing *UAS-htl^{λ}* in clones, we observed a high percentage of ectopic mng in the cortex at 24 and 42 h APF (43 and 75%) that increased to 81% in adults (Fig. 8d–g). Thus, clonal expression of *htl^{λ}* provided us with a tool to generate ectopic mng in the cortex and to test the contribution of *lapsyn* in regulating survival following abnormal positioning. We observed that *lapsyn* loss in *UAS-htl^{λ}* expressing

clones did not alter the percentage of ectopic mng in the cortex. This suggests that FGF signaling but not *lapsyn* controls mng survival. When we co-expressed *UAS-lapsyn* and *UAS-htl^λ*, the percentage of mng in the cortex was reduced by 53% (Supplementary Fig. 7). This further supported the notion that the role of *lapsyn* in anchoring mng, possibly by counter-acting the pro-migratory role of FGF signaling, is pivotal. Surviving *htl^λ*-expressing mng formed extensive stellate branches in the cortex. Moreover these, as well as mng at the neuropil border, extended short tufted arbors into superficial medulla neuropil layers. mng clones that expressed *UAS-htl^λ* but lacked *lapsyn* showed substantially less branching in the cortex and in the neuropil

compared to clones overexpressing *UAS-htl^λ* (Fig. 8e–f). Together, these findings support the model (Supplementary Fig. 8) that *lapsyn* mediates mng anchorage at the neuropil border, and thus indirectly survival, which depends on gliotrophic FGF signaling. Moreover, *lapsyn* acts downstream of or in parallel with FGF signaling to control branch morphogenesis of astrocyte-like mng, revealing the robust control of this developmental step through independent molecular mechanisms.

**Requirement of *lapsyn* in other CNS regions.** Finally, we asked whether *lapsyn* mediates astrocyte branch morphogenesis beyond the visual system. The *lapsyn^fTRG027706-135B* fosmid reported protein expression in astrocyte populations in the adult antennal lobe, the central brain, and mushroom body lobes, as well as in the second and third instar larval and adult VNC (Fig. 9a, d, g and Supplementary Fig. 9a, b). *loco^1.3D2*-Gal4 was active in all regions examined and therefore suitable for driving *UAS-lapsyn^IR-KK102333* expression (Fig. 9b, e, h and Supplementary Fig. 9c). Because astrocytic branches in mushroom body lobes were sparse, effects following *lapsyn* knockdown in astrocyte-like glia were less conspicuous in these neuropils. However, branch morphogenesis in the antennal lobes, the central brain and adult and third instar larval VNC was severely reduced (Fig. 9c, f, i and Supplementary Fig. 9d). Together, these findings indicate that *lapsyn* plays a general role in regulating astrocyte branch formation in the *Drosophila* CNS regardless of their developmental origins.

## Discussion

Vertebrate astrocytes in different brain regions are considered as developmentally and functionally diverse cell types[2, 12, 13]. In *Drosophila*, recent studies elucidated the development and function of astrocyte-like glia in the VNC, mushroom bodies, and antennal lobes[5, 6, 16, 21, 22, 26]. Our characterization of mng in the visual system further supports the notion that *Drosophila* astrocyte-like glia constitute a heterogeneous population with distinct origins, developmental behaviors, and morphologies. Larval VNC astrocyte-like glia are derived from embryonic longitudinal glioblasts, while their counterparts in the central brain originate from embryonic glioblasts and dedicated post-embryonic type II neuroblasts. These astrocyte subtypes expand by mitotic divisions as they migrate over neuropil surfaces[52, 53]. By contrast, OPC neuroepithelial-derived neuroblasts mature into neuroglioblasts to generate astrocyte-like mng that rarely divide[40, 41] (this study). During early metamorphosis VNC and brain astrocyte-like glia lose their processes and transform into phagocytes that engulf synapses and neuronal fragments before regrowing branches during late pupal development[8] or are

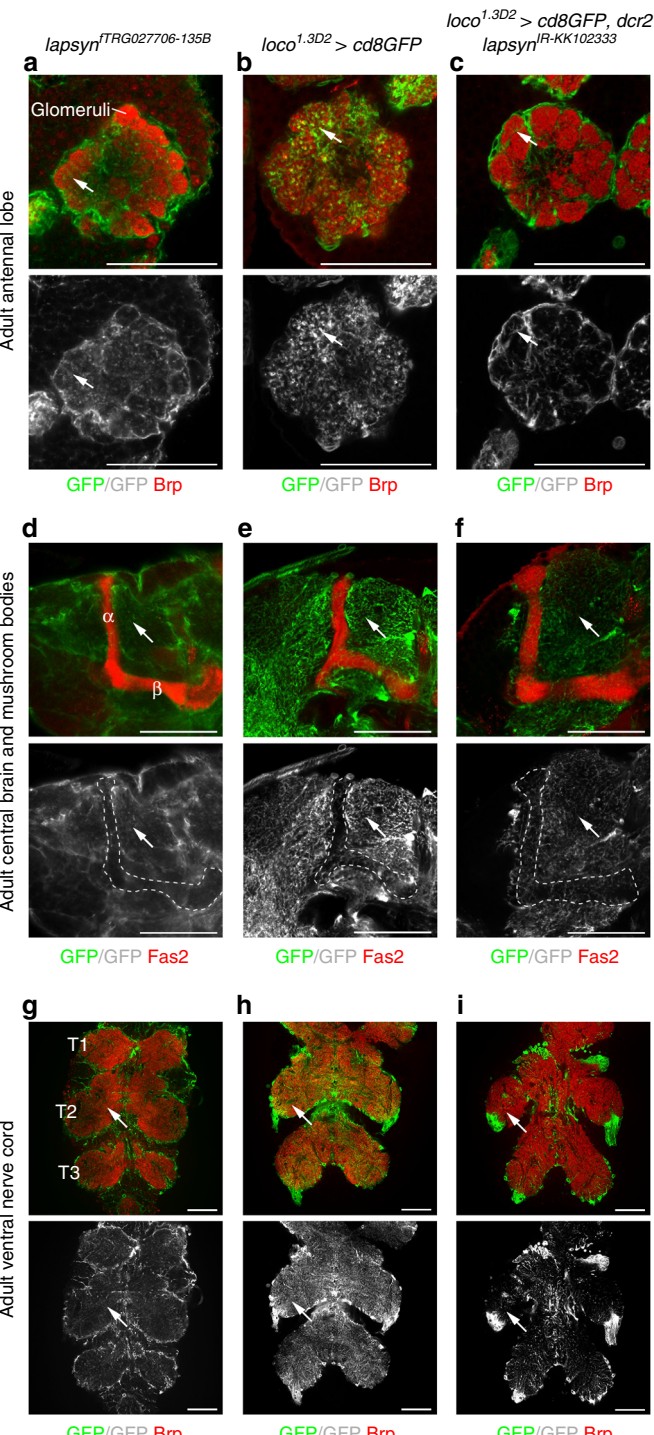

**Fig. 9** *lapsyn* is widely required for branch morphogenesis of astrocyte-like glia in the CNS. Brp was used to visualize synaptic neuropils (*red*, **a–c**, **g–i**). Fas2 labeled the mushroom body lobes (**d–f**, *red*). **a**, **d**, **g** The fosmid *lapsyn^fTRG027706* reported lapsyn expression (*green*, arrows) in adult astrocyte-like glia extending into glomeruli within the antennal lobe (**a**), the neuropils of the central brain (**c**), and the VNC (**g**) in addition to other glia. **b**, **e**, **h** *loco^1.3D2*-Gal4 UAS-cd8GFP (*green*) labeled branches of astrocyte-like glia in all examined CNS areas. In mushroom body α and β lobes (**d**, **e**, **f**, *outlined*), lapsyn expression was weak, in line with the low density of astrocyte-like glial branches labeled with *loco^1.3D2*-Gal4 UAS-cd8GFP, limiting assessment of defects following *lapsyn* knockdown. Compared to controls **b**, **e**, **h**, the knockdown of *lapsyn* in astrocyte-like glia resulted in reduced branch morphogenesis (arrows) in the adult antennal lobe, the central brain and VNC (**c**, **f**, **i**). All panels show single optical sections. For genotypes and sample numbers, see Supplementary Table 1. Scale bars, 50 μm

removed by apoptosis[53]. However, astrocyte-like mng—similar to their antennal lobe counterparts[5]—associate with a newly arising, initially synapse-free neuropil. Interestingly, antennal and optic lobe astrocyte-like glia extend processes in a similar time window that coincides with the onset of synaptogenesis[5]. Finally, unlike the asymmetric stellate VNC and brain astrocyte-like glia, dmng variants are elongated and associate with distinct medulla neuropil layers. Their stereotypic morphology argues that branch formation is controlled by layered and intracolumnar patterning mechanisms, which may include differential adhesive interactions.

Extracellular LRR superfamily members control axonal and dendritic targeting, as well as synapse formation and plasticity of neurons[54]. Our expression and functional analyses in *Drosophila* uncovered a central role for the LRR transmembrane protein Lapsyn[33] in regulating the development of astrocyte-like glia. Compared to neuronal LRR proteins, only very few members such as the secreted astrocytic leucine-rich repeat molecule (Alrm) in *Drosophila* and Oligodendrocyte-myelinating glycoprotein (OMgp) in vertebrates have been associated with glia[16, 54]. Tandem arrays of LRR motifs form arc or horseshoe-shaped structures. These serve as versatile protein recognition motifs that promote cell-cell interactions by binding to specific, albeit structurally unrelated ligands[54, 55]. Each repeat consists of 20–30 residues subdivided into a highly conserved and a variable segment, whose consensus sequences and length help to sub-divide LRR motifs into classes[55]. LRR proteins underwent sub-stantial expansion and diversification in different animal species[45]. Lapsyn contains no other recognizable motifs than the LRR and transmembrane domains. In classifications based on the presence of non-LRR domains, unrelated members lacking any other motifs tend to group. However, categorizations assessing arrays based on LRR class composition helped to cluster proteins with potentially similar functional structures[56]. In line with pre-vious reports[33], our searches using SmartBLAST and CATH/ Gene3D programs[57] predicted leucine-rich repeat-containing protein 4 (LRRC4), also known as Netrin-G ligand (NGL), family members as remotely related proteins of Lapsyn in vertebrates with 29% identity. LRRC4C (NGL-1), LRRC4 (NGL-2), and LRRC4B (NGL-3) contain one immunoglobulin domain in addition to the LRR chain, and regulate neurite outgrowth, laminar targeting and synapse formation[58]. Interestingly, NGL-2 also has been linked to astrocyte differentiation and glioma[59]. While NGL-1 and NGL-2 bind to mammalian-specific Netrin-G1 and 2, NGL-3 interacts via the LRR domain with the highly conserved leukocyte-antigen-related-like (LAR) and other protein tyrosine phosphatases[60], underscoring the diversity of non-conserved and conserved binding partners for this LRR sub-family. However, to further elucidate Lapsyn function and its possible link to NGL family members, it is essential to identify its molecular partners using unbiased approaches in the future.

*lapsyn* is expressed in astrocyte-like mng and other glia, but not in neurons as previously described[61]. The tiled organization and minimal overlap of processes in adult mng suggests that—similarly to VNC glia[26]—branches stop growing once they infiltrated the entire neuropil and come in contact with those of neighboring glia. Because emerging processes from adjacent wild-type astrocyte-like mng barely touch and loss of *lapsyn* negatively affects process extension, this cell surface molecule is unlikely to mediate contact-dependent homotypic repulsive interactions. While astrocyte-like and ensheathing mng cell bodies are inter-spersed at the neuropil border, the branches of astrocyte-like mng are primarily in contact with neurons and only slightly with the sparse branches of ensheathing mng. It is therefore conceivable that Lapsyn is part of a signaling complex in astrocyte-like mng membranes that promotes cell-cell recognition events or adhesive interactions with as yet elusive heterotypic neuronal ligands to enable anchorage at the neuropil border and branch elaboration within the neuropil. At present, we cannot exclude additional interactions with ensheathing mng for positioning and initial branch extension, or an independent *lapsyn* requirement in this glial subtype. Persistent clonal overexpression of *lapsyn* did not affect astrocyte-like mng morphology, suggesting that this determinant does not promote growth and the tight temporal regulation of expression levels is not essential for its function.

Different from other CNS areas, some astrocyte-like mng leave the neuropil border and migrate into cortex during early pupal development, where they subsequently undergo apoptosis. Consistently, the number of wild-type mng clones decreases during metamorphosis. The medulla and lobula complex undergo a 90° rotation with respect to the lamina[62], and medulla neuron somata extensively relocate to new positions in the cortex during early metamorphosis[63]. Therefore, mng find themselves in an environment particularly conducive to cell migration. Additionally, mng may compete for space and potential survival factors at the neuropil border, and excess cells may become expelled. Thus, *lapsyn* may be part of a mechanism ensuring that sufficient numbers of astrocyte-like mng are retained at the neuropil border.

The motility and survival of astrocyte-like mng depend on FGF signaling levels. Increased FGF receptor activity causes mng to relocate from the neuropil border into the cortex, consistent with the well-established pro-migratory role for this pathway[50]. Early *htl* knockdown reduced, while over-activation increased mng numbers. Importantly, in addition to some increase in pro-liferation, the percentage of apoptotic $htl^\lambda$ expressing astrocyte-like mng in the pupal medulla cortex was reduced and ectopic cells persisted into adulthood. This argues that *htl* signaling regulates survival. This less explored function of FGF signaling[50] could either be primary or involve secondary pathways such as epidermal growth factor receptoractivation[64].

Higher levels of Htl signaling increase the territories occupied by VNC astrocyte clones[26]. By contrast, enhanced Htl signaling in mng clones results in the formation of tufted arbors with short densely packed branches that are unable to extend into deeper neuropil layers. Hence, *htl* in mng may be required for promoting branch growth rather than acquiring specific patterns. *lapsyn* overexpression fails to rescue branch extension defects or cell loss caused by *htl* knockdown, suggesting that FGF signaling is not epistatic to *lapsyn*. However, mng—overexpressing constitutively active Htl while lacking *lapsyn*—survive into adulthood and dis-play severely reduced branches. These genetic data support the notion that *lapsyn* acts downstream of or in parallel with FGF signaling to regulate branch formation, whereas FGF signaling, but not *lapsyn*, mediates survival. *lapsyn* overexpression is likely insufficient to rescue *htl* knockdown because additional factors may be affected that govern robust glial branch morphogenesis. Higher *lapsyn* levels reduced $htl^\lambda$-induced migration of mng into the cortex. The differential effects on branch extension and positioning argue that *lapsyn* may not simply enhance or attenuate FGF signaling.

The neuron-derived FGF8-like ligand Ths promotes ensheathment of R-cell axon bundles by wrapping glia in the larval eye imaginal disc[25] and astrocyte branch extension into the VNC neuropil[26]. Similarly, we observed that Ths is produced by neurons in the optic lobe. Together, this suggests a model, whereby parallel signals control the balanced development of astrocyte-like mng: neuronal FGF could function as a gliotrophic factor, and promote motility and branch growth, whereas other as yet unidentified neuron-derived factors could interact with Lapsyn to achieve the correct extension and ramification of processes within the neuropil. These or additional ligands may

also mediate neuropil recognition and anchorage of mng at the neuropil border.

In summary, we identified potentially related roles for the transmembrane protein Lapsyn in regulating branch extension and positioning of astrocyte-like mng. Their complex adult morphology indicates that further factors await discovery to explain stepwise branch formation at astrocyte-neuron interfaces. Because astrocytes have been linked to several neurodevelopmental disorders in humans[11], this detailed knowledge could ultimately contribute to our better understanding of underlying disease mechanisms.

## Methods

**Genetics.** *Drosophila melanogaster* strains were maintained on standard medium at 25 °C, if not specified otherwise. The following stocks or crosses were used to label different glial cell subtypes: (1) repo-Gal4, (2) R56F03-Gal4[32, 35, 36], (3) NP6520-Gal4[17], (4) loco[1.3D2]-Gal4[37], (4) second and third chromosomal alrm-Gal4 insertions[16] (by M. Freeman, University of Massachusetts), and (5) dEAAT1-Gal4 (ref. [18]; by S. Birman, ESPCI) were crossed to UAS-cd8GFP. A homozygous loco[1.3D2]-Gal4; UAS-cd8GFP stock was used in some experiments. The stock GMR-myr mRFP; loco[1.3D2]-Gal4; UAS-cd8GFP, UAS-dcr2 was used to genetically label glia and R-cell axons in the same sample. Individual glial cells were visualized using the Flybow approach[31] by crossing loco[1.3D2]-Gal4; hs-mFLP5/TM6B to UAS-FB1.1[260b]. Progeny were subjected to two 45 min heat shocks in a 37 °C water bath at 48 and 72 h after egg laying (AEL) and dissected during indicated stages. To visualize ensheathing and astrocyte-like glia in the same sample, the stock UAS-FB1.1B[260b]; loco[1.3]-lexA lexAop-myr mCherry was crossed to R56F03-Gal4. The loco[1.3]-lexA insertion on the third chromosome was generated for this study (see below).

Lineage analysis experiments were performed using MARCM[42] by crossing (1) FRT19A tubP-Gal80 hs-FLP[1]; UAS-lacZ[nls] UAS-cd8GFP; tub-Gal4/TM6B to (2) FRT19A/Y. To induce clones in the optic lobe, 12 h embryo collections were heat shocked for 10–20 min at 60–72 h AEL in a 37 °C water bath. Third instar larval brains were dissected during the following 24–48 h. MARCM clones were generated using the ubiquitous tubP-Gal4 driver and hs-FLP as a recombinase source to label all cell types with GFP in a clone. Other clone configurations than those described in Fig. 2b–d that included mng were not detected, supporting the notion that OPC neuroblasts maturing into neuroglioblasts are the primary source for mng in the medulla.

For the RNAi-based screen, loco[1.3D2]-Gal4; UAS-cd8GFP, UAS-dcr2 flies were crossed to 522 UAS-RNAi lines obtained from the Vienna Drosophila Resource Center (VDRC)[65] and National Institutes of Genetics (Japan) collections (http://www.shigen.nig.ac.fly/nigfly). We focused on lines affecting genes that encode secreted and cell surface proteins with potential expression or functions in the nervous system. Candidates were selected based on information provided in refs. [45, 66, 67]. Progeny of crosses were raised either at 29 °C or if lethal at 25 °C. UAS-RNAi lines that caused lethality under either condition were retested in a second screen, in which heat shock-induced FLP (hs-FLP) led to mosaic expression by excision of the FRT site flanked Gal4-repressor Gal80 downstream of a widely active tubulin enhancer[68] (tub>Gal80>). Specifically, RNAi lines were combined with hs-FLP[122], tub>Gal80>; loco[1.3D2]-Gal4; UAS-cd8GFP, UAS-dcr2. Progeny were shifted to 29 °C at 24–48 h AEL and exposed to a 10-min heat shock at 37 °C between 48 and 72 h AEL to achieve a spatially and temporally restricted knockdown. Lapsyn knockdown in astrocyte-like glia was achieved by crossing loco[1.3D2]-Gal4; UAS-cd8GFP, UAS-dcr2 to (1) UAS-lapsyn[IR-KK102333] and (2) UAS-lapsyn[IR-15658R1] or by crossing loco[1.3D2]-Gal4; UAS-cd8GFP to UAS-lapsyn[IR-KK102333]; UAS-dcr2. Progeny were shifted to 29 °C at 24–48 h AEL to achieve optimal knockdown.

loco[1.3D2]-Gal4, UAS-cd8GFP was combined with UAS-ChRFP-Tub to visualize microtubules. loco[1.3D2]-Gal4,UAS-cd8mCherry[260b] transgenes were combined with (1) UAS-Eb1-GFP to track plus-end microtubules, (2) UAS-nod-GFP to label minus-end microtubules, and (3) UAS-LifeAct-GFP to visualize actin. To assess the effect of lapsyn knockdown on the cytoskeleton, UAS-lapsyn[IR-KK102333]; UAS-dcr2 was crossed to (1) loco[1.3D2]-Gal4, UAS-cd8mCherry[260b]; UAS-Eb1-GFP and (2) loco[1.3D2]-Gal4, UAS-cd8mCherry[260b]; UAS-LifeAct-GFP.

To identify the cell types producing Lapsyn, optic lobes of heterozygous Mi{MIC}lapsyn[MI01316]/CyO progeny[46] were analyzed. Lapsyn protein expression was assessed in Mi{MIC}lapsyn[lapsyn2xHA-24A] and Mi{MIC}lapsyn[lapsyn2xHA-73B] transgenic flies, generated by injection-based RMCE. Furthermore, endogenous Lapsyn protein expression was determined in fly lines containing the Crispr/Cas9 allele lapsyn[2xHA-CR13.1] and the lapsyn[fTRG027706-135B] fosmid[48] (see below). The former were crossed to loco[1.3D2]-Gal4; UAS-cd8GFP. Lapsyn localization in glial processes in gain-of-function studies was assessed by crossing loco[1.3D2]-Gal4 to UAS-lapsyn-GFP (ref. [33]; by T. Littleton (Massachusetts Institute of Technology)).

For whole animal neuron- and glial cell-specific rescue experiments, lapsyn[ZG1], a null allele carrying a ~4.7 kb deletion of the 5′UTR and the first two coding exons was used[33] (by T. Littleton). (1) elav[C155]-Gal4; lapsyn[ZG1]/CyO Dfd-YFP and (2) lapsyn[ZG1]/CyO Dfd-YFP; repo-Gal4/TM6B were crossed to lapsyn[ZG1]

/CyO Dfd-YFP; UAS-lapsyn. Rescue experiments were conducted in independent triplicates, using overnight grape juice agar plate collections of a total of 1210 and 2360 embryos, respectively. Rescue of viability at the first instar larval stage was calculated as percentage of expected progeny. Subsequently, hatched first instar larvae in "rescue" and "control" populations from above egg collections were monitored for survival as third instar larvae, as young (0–24 h APF) and old pupae with dark appearing wings development, and as adults. "Rescue" animals have the following genotypes: elav[C155]-Gal4; lapsyn[ZG1]/lapsyn[ZG1]; UAS-lapsyn/+ and lapsyn[ZG1]/lapsyn[ZG1]; repo-Gal4/UAS-lapsyn. "Control" animals included all genotypes that were heterozygous for lapsyn[ZG1], or heterozygous for lapsyn[ZG1] while overexpressing lapsyn. The UAS-lapsyn transgene[33] was generated by T. Littleton. The lapsyn[fTRG027706-135B] transgene reported Lapsyn expression in second and third instar larval VNCs solely in glia. elav-Gal4[C155] is known to be transiently active in glia during early development[69]. These low levels may enable very unhealthy animals to survive during early larval development.

To conduct loss- and gain-of-function, as well as rescue experiments in clones, the ey-FLP transgene was combined with MARCM (ey-FLP MARCM): ey-FLP; FRT42B tubP-Gal80 was crossed to (1) loco[1.3D2]-Gal4, FRT42B; UAS-cd8GFP (control), as well as (2) loco[1.3D2]-Gal4, FRT42B lapsyn[ZG1]/CyO Dfd-YFP; UAS-cd8GFP, and (3) loco[1.3D2]-Gal4, FRT42B lapsyn[LL00906]/CyO Dfd-YFP; UAS-cd8GFP (lapsyn loss-of-function); ey-FLP; FRT42B tubP-Gal80; UAS-lapsyn/TM6B was crossed to (1) loco[1.3D2]-Gal4, FRT42B; UAS-cd8GFP (gain-of-function) and (2) loco[1.3D2]-Gal4, FRT42B lapsyn[ZG1]/CyO, Dfd-YFP; UAS-cd8GFP (rescue in astrocyte-like mng clones). For overexpression in all astrocyte-like mng, loco[1.3D2]-Gal4; UAS-cd8GFP was crossed to UAS-lapsyn. lapsyn[LL00906] is a mutagenic piggyBac element inserted into the lapsyn locus 260 bp after the transcriptional start site on a chromosome also carrying a FRT42B site[49]. While we observed qualitatively similar defects for lapsyn[LL00906] and lapsyn[ZG1], the penetrance was slightly lower for lapsyn[LL00906]. For lapsyn knockdown and overexpression in medulla cortex glia, wrapper[GMR54H02]-Gal4[35] was recombined with UAS-myrmRFP and subsequently crossed to (1) UAS-lapsyn[IR-KK102333]; UAS-dcr2 and (2) UAS-lapsyn. For rescue experiments by overexpression of p35, ey-FLP; FRT42B tubP-Gal80; UAS-p35 was crossed to (1) loco[1.3D2]-Gal4, FRT42B; UAS-cd8GFP (control) and (2) loco[1.3D2]-Gal4, FRT42B lapsyn[ZG1]/CyO Dfd-YFP; UAS-cd8GFP.

To visualize control or mutant clones and surrounding heterozygous astrocyte-like mng, ey-FLP; FRT42B tubP-Gal80; loco[1.3]-lexA lexAop-myr mCherry/TM6B was crossed to (1) loco[1.3D2]-Gal4, FRT42B; UAS-cd8GFP and (2) loco[1.3D2]-Gal4, FRT42B lapsyn[ZG1]/CyO Dfd-YFP; UAS-cd8GFP.

For expression analysis of FGF receptor pathway components, (1) htl[fTRG482] (ref. [48]), (2) wild type[OreR] and (3) Mi{MIC}ths[MI01564] were used. To assess interactions with the FGF signaling pathway, loco[1.3D2]-Gal4; UAS-cd8GFP was crossed to (1) UAS-htl[IR-GD6692] and (2) UAS-htl[IR-GD27180] (knockdown; refs. [26, 65]), (3) UAS-htl[IR-GD27180]; UAS-lapsyn (rescue) and (4) UAS-htl[λ] (over-activation; ref. [51]). Additionally, GMR-myr RFP; loco[1.3D2]-Gal4; UAS-cd8GFP, UAS-dcr2 was crossed to UAS-htl[IR-GD27180]. Because early htl knockdown by continuous expression at 29 °C severely interfered with the survival of astrocyte-like mng, progeny were maintained at 18 °C until 42 h APF (equivalent to 21 h APF at 25 °C) and then transferred to 29 °C. To increase FGF signaling in wild type or lapsyn[ZG1] astrocyte-like mng clones, ey-FLP; FRT42B tubP-Gal80; UAS-htl[λ]/TM6B was crossed to (1) loco[1.3D2]-Gal4, FRT42B; UAS-cd8GFP and (2) loco[1.3D2]-Gal4, FRT42B lapsyn[ZG1]/CyO Dfd-YFP; UAS-cd8GFP. To increase FGF signaling in wild type or UAS-lapsyn overexpressing astrocyte-like mng, loco[1.3D2]-Gal4; UAS-cd8GFP was crossed to (1) UAS-htl[λ] and (2) UAS-htl[λ], UAS-lapsyn. Supplementary Table 3 and references provide additional information for Drosophila stocks used in this study. If not otherwise indicated, lines were obtained from the Bloomington Drosophila Stock Center and assembled into above described stocks. Generally, crosses involved about 5–8 males and 10–15 unfertilized females. To avoid overcrowding, parents were transferred to fresh vials every day for crosses at 25 °C and every three days at 18 °C.

**Molecular biology.** To generate the lapsyn[2xHA] construct for insertion into the attP site containing Mi{MIC}lapsyn[MI01316] line using φC31 integrase-mediated RMCE by injection[46], the lapsyn ORF was PCR amplified from the Gold collection clone GH22922 (Drosophila Genomics Resource Center) with the forward primer lapsyn[F1] and the reverse primer lapsyn[R2] 5′-GAGGCTAGCGGCGTAATCGGG CACATCGTAGGGGTATTTTCGGGCAATATCACAGTG-3′, that added a NheI site and the first HA-tag encoding sequence. This fragment was subcloned into the pTV vector[70] using EcoRI and NheI restriction enzyme sites. The lapsyn 3′UTR was PCR amplified using the forward primer lapsyn[F2] 5′-AAGCTAGCTACCCCTAC GATGTGCCCGATTACGCCTAGCACTTTAGTTAGTTAATTAGTTGCTTAG-3′ that added a NheI site and a second HA-tag encoding sequence, as well as the reverse lapsyn[R1] primer. This fragment was inserted into the pTV vector containing the lapsyn-HA fragment using NheI and BamHI restriction enzyme sites. The full-length lapsyn[2xHA]3′UTR construct was subcloned into pBS-KS-attB1-2-GT-SA (Addgene; ref. [46]) using EcoRI and BamHI sites. Injections and crosses to generate Mi{MIC}lapsyn[lapsyn2xHA] and alleles were performed as described in ref. [46]. Testing resulting lines for the correct orientation of RMCE events, we identified two lines (Mi{MIC}lapsyn[lapsyn2xHA-24A] and Mi{MIC}lapsyn[lapsyn2xHA 73B]) as correct insertions with the help of the following primers: RMCE[1] 5′-TAGAGTGATGTGT GTGTAGTGAACTAAC-3′, RMCE[A] 5′-CGGAAGAGAGATAAATCGGTTG-3′,

*RMCE*[B2] 5′-GCCCAGAAACGCCATCAAC-3′, and *RMCE*[2.2] 5′-TCCCTATTC TCGAGGCTTTGAT-3′.

To generate the *lapsyn*[2xHA-CR13.1] allele, a PAM site and guide sequence closest to the *lapsyn* stop codon were selected (Fig. 4). Forward and reverse oligonucleotides of the target sequence (Oligo[top] 5′-gtcGACTAACTAAAGTGCTA TTTT-3′ and Oligo[bottom] 5′-aaacAAAATAGCACTTTAGTTAGT-3′) with *BbsI*-compatible ends were annealed and ligated into the *pCFD3* plasmid digested with *BbsI* (ref. [47]; by S. Bullock (MRC LMB) and C. Alexandre (Francis Crick Institute)). A single-stranded oligo donor nucleotide (ssODN, 169-nt) that contained two 50-nt homology arms flanking the sequences of two HA epitope tags separated by a GGGS spacer, was synthesized by IDT (Integrated DNA Technologies): 5′-ACAACCGTCAGCCGGAGGACGAGCCTCTGCACTGTGAT ATTGCCCGAAAATACCCATACGACGTCCCTGACTATGCGGGAGGTGGGA GTTATCCCTATGATG TGCCCGATTACGCTTAGCACTTTAGTTAGTTAAT TAGTTGCTTAGTTAGTTAGTTAGTCTTAGGTGC-3′. The ssODN (250 ng/μl) and *pCFD3* plasmid (300 ng/μl) were co-injected into *y*[1]*M{vasa-Cas9.RFP-}ZH-2A w*[1118] (BDSC 55821) embryos. Thirty-five viable and fertile recovered male and female flies were crossed individually to 3–4 *yw; GlaBc/CyO* female or male flies, respectively. In the next generation, 1–8 male offspring from each vial were crossed individually to 3–4 *yw; Gla Bc/CyO* females. Once progeny were visible, these males were sacrificed to prepare genomic DNA for PCR screening using the following primers: VasC9 seq2F: 5′-GACGAAAGAGGAATGCCCAGAAG-3′ and VasC9 seq2R: 5′-GGTAAAGGATTCAAGGCAGTTTG-3′. In 2 of 131 candidates, the HA epitope tag was inserted in frame upstream of the *lapsyn* stop codon. The line *lapsyn*[2xHA-CR13.1] was used for further analysis.

The tagged fosmid *lapsyn*[fTRG027706] clone (Source BioScience) was amplified and sequence-verified following the protocol of Sarov et al.[48]. The fosmid (250 ng/μl) was injected into *y*[1]*w P{nos- ϕC31\int.NLS}X; PBac{y*[+]*-attP-3B} VK00033* flies (BDSC: 32542). We recovered two independent insertions on the third chromosome. The line *lapsyn*[fTRG027706-135B] was used for further analysis.

To generate *loco*[1.3]*-LexA*, the 2.2 kb *"1.3" loco* enhancer fragment was removed from the *p221-4* plasmid (ref. [37]; by C. Klämbt (University of Münster)) and inserted into the Gateway® selection vector *pENTR 1A* (Invitrogen™) using *Eco*RI restriction enzyme sites. This fragment was subcloned into the plasmid *pBPLexA:: p65Uw* (Addgene #26231) using standard Gateway cloning technology. Transgenic flies with insertions in the *attP2* site (68A4) on the third chromosome were generated by Rainbow Transgenic Flies, Inc. (Camarillo, CA, USA) using a ϕC31 integrase-based injection protocol.

**Immunolabeling and imaging.** Brains of larvae, pupae, and pharate adults were dissected in phosphate-buffered saline (PBS), fixed for 1 h at 20–24 °C in 2% paraformaldehyde (wt/vol) in 0.05 M sodium phosphate buffer (pH 7.4) containing 0.1 M L-lysine (Sigma-Aldrich) and washed in PBT (PBS containing 0.5% Triton X-100 (Sigma-Aldrich)). The following primary antibodies were diluted in 10% Normal Goat Serum (NGS) and PBT for immunolabeling over night at 4 °C: mouse antibody mAb24B10 (1:75, Developmental Studies Hybridoma Bank (DSHB)), mouse anti-Brp (1:10, DSHB nc82), rat antibody CadN (1:20, DSHBDN-Ex #8), rabbit antibody to Dcp1 (1:200, Cell Signaling Technology #9578), rabbit antibody to dEAAT1(1:5000, by D. Van Meyel, McGill University, ref. [36]), rabbit antibody to Dof (1:200, shared by C. Klaembt, University of Münster)) and rabbit antibody to Fas2 (1:10, DSHB, 1D4), rabbit antibody to GAT (1:2000, by M. Freeman, ref. [26]), rabbit antibody to GFP (1:1000, Invitrogen A6455), rabbit antibody to PH3 (1:100, Millipore/Upstate #06-570), rabbit antibody to Gs2 (1:100, Merck Millipore MAB302), mouse antibody to Prospero (1:50, DSHB MR1A), mouse antibody to Repo (1:20, DSHB, 8D12), rabbit antibody to Repo (1:500, by J. Urban, University of Mainz, ref. [34]) and rabbit antibody to RFP (1:500, Abcam ab62341). For immunofluorescence labeling, samples were incubated in the following secondary antibodies for 2.5 h at 20–24 °C: goat anti-mouse, anti-rabbit or anti-rat F(ab′)₂ fragments coupled to Cy3 or Alexa Fluor®647 (1:400; Jackson ImmunoResearch Laboratories; antibody to mouse: Cy3 (115-166-003), Alexa Fluor®647 (115-606-003); antibody to rabbit: Cy3 (111-166-003), Alexa Fluor®647 (111-606-003); antibody to rat: Cy3 (112-166-003)). For immunolabeling of *Mi{MIC}lapsyn*[lapsyn2xHA-24A] optic lobes, over-night incubations were performed with primary rabbit antibody to HA-Tag (1:300, Cell Signaling Technology C29F4) and with secondary goat antibody to rabbit: Cy3 (111-166-003) for 4 h at 20–24 °C. For HA-immunolabeling of *lapsyn*[2xHA-CR13.1] optic lobes, following the incubation in primary antibody over night at 4 °C, samples were washed for 15 min in PBT at 20–24 °C and overnight in NGS/PBT at 4 °C, and then transferred into secondary goat antibody to rabbit: Alexa Fluor®555 (Life Technologies, A21430) for 2 h at 20–24 °C. Images were collected using a Leica TCS SP5II laser scanning confocal microscope. Adobe Photoshop, Fiji and Volocity (Perkin Elmer) software programs were used for image analysis. mng clones are represented as projections of several optical sections to show their full extent of processes.

**Quantifications.** To quantify defects in glial branch morphogenesis, localization and survival, optic lobes of control and experimental animals were imaged in a horizontal orientation. GFP-positive astrocyte-like mng were assessed in one to three independent 20-μm stacks of optical sections per sample for all MARCM experiments. In quantifications shown in Figs. 7h and 8c, single optical sections were examined. In quantifications shown in Fig. 7j, l, 2-μm-image stacks of entire optic lobes were analyzed. Lateral, proximal, and distal astrocyte-like mng were assessed based on the position of their cell bodies relative to the neuropil with the exception of optic lobes at 24 h APF, when glial cells have not yet migrated to their final position and therefore were all considered as one group. Furthermore, positioning of astrocyte-like mng in the cortex or at the neuropil border was determined based on the following criteria. Cells were considered to be located at the neuropil border if part of the cell body (i) touches the neuropil labeled with an antibody against N-cadherin (CadN), (ii) is in close contact with other mng at the neuropil border, or (iii) is touching R-cell axons at 24 h APF. Otherwise, cells were considered to be localized in the medulla cortex regardless of long thin processes contacting the neuropil border. SmartBLAST and CATH/Gene3D[57] programs were used to search for remote vertebrate homologs of *Drosophila* Lapsyn.

**Statistics.** In a pool of control or experimental animals, flies with unambiguously identifiable genotypes were selected randomly and independently from different vials at defined developmental stages. Data acquisition and analysis were not performed blindly because of the reliance on defined genotypes. Samples sizes were not pre-calculated using statistical tests but followed the standard of the field. Statistical calculations were performed using Prism 6 GraphPad and Microsoft Excel software. D'Agostino-Pearson omnibus normality tests indicated that data sets generally met the assumption for normality. *P* values were determined using the parametric unpaired two-tailed Student's *t*-test (Welch corrected, not assuming equal variance or standard deviations). If a small set of data in a larger experiment did not pass this test (owing to few outlier data points or too small n numbers), normality was assumed and parametric Student's *t*-tests were conducted. If the majority of data subsets did not meet the assumption of normality, non-parametric, two-tailed Mann-Whitney *U*-tests, corrected for ties, were performed. Quantifications are presented as stacked 100% bar graphs and as scatter plots and bar graphs. Data are presented as means ± 95% confidence interval error bars or as means ± standard deviation interval error bars. *$P < 0.05$; **$P < 0.01$; ***$P < 0.001$; ****$P < 0.0001$. T values (t) and degrees of freedom (df), as well as *U* values are listed in Supplementary Table 2.

**Data availability.** Image data sets generated and analyzed in this study are available from the corresponding author on reasonable request. A source data file for quantifications shown in Figs. 2, 4–8, and Supplementary Fig. 7 is provided with the paper (Supplementary Data 1).

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

## Acknowledgements

We are grateful to T. Littleton (Massachusetts Institute of Technology), C. Klämbt (University of Münster), M. Freeman (University of Massachusetts), D. Van Meyel (McGill University), S. Birman (ESPCI), B. Stramer (King's College London), N. Tapon (Francis Crick Institute), S. Bullock (MRC LMB), and C. Alexandre (Francis Crick Institute) for sharing fly strains, antibodies, and constructs. We thank the Bloomington *Drosophila* Stock Center (NIH P40OD018537), the Vienna *Drosophila* Resource Center, the Japan National Institutes of Genetics fly stock center, the Kyoto *Drosophila* Genetic Resource Center, and the Developmental Studies Hybridoma Bank for fly strains and antibodies. We thank M. Pilatova and H. Apitz (Francis Crick Institute) for assistance with the verification and immunostaining of *Mi{MIC}lapsyn*$^{lapsyn2xHA}$, E. Powell (Francis Crick Institute) for the *loco*$^{1.3D2}$-*Gal4 UAS-cd8mCherry*$^{260b}$ recombinant, C. Alexandre for the protocol and guidance with Crispr/Cas9 experiments, and W. Taylor (Francis Crick Institute) for advice on homology searches. We thank J.P. Vincent, F. Guillemot, H. Apitz, K. Dolan, R. Kaschula, and N. Shimosako (Francis Crick Institute), as well as E. Ober (DanStem) and B. Hofbauer (University of Würzburg) for critical reading of the manuscript. This work was supported by a fellowship of the Heinrich Hertz-Stiftung (S.M.), the Medical Research Council (U117581332 I.S.), and the Francis Crick Institute (FC001151 I.S.), which receives its core funding from Cancer Research UK, the Medical Research Council and the Wellcome Trust.

## Author contributions

B.R. and I.S. conceived the project. B.R. and C.D.V. performed the experiments and analyzed the data. S.M. contributed to the *lapsyn* knockdown and *lapsyn*$^{LL00906}$ mosaic analyses experiments. I.S. contributed to the characterization of the FGFR signaling pathway. I.S. wrote the manuscript with support of all authors.

## Additional information

**Competing interests:** The authors declare no competing financial interests.

