## [Peer Review File · Nature Communications]

Reviewers' expertise:

Reviewer #1: Drosophila, neuro-glia interaction;

Reviewer #2: Drosophila, neuro-glia interaction, cell morphogenesis;

Reviewer #3: Drosophila, developmental signaling.

Reviewers' comments:

Reviewer #1 (Remarks to the Author):

This is a very nice study from Salecker and colleagues that explores the development of medulla neuropil glia (mng). The authors describe the detailed morphology of these cells, developmental elaboration, cell-cell interactions with other mngs and neurons, and timing of infiltration relative to synaptogenesis. Most interestingly, through a forward genetic screen they identify Lapsyn, and LRR molecule required for mng growth and survival. They explore the timing of expression and function of Lapsyn, its roles in mng survival, and genetic interactions with FGF signaling. The authors should be commended for rigorous genetics, beautiful figures, and for establishing an interesting system in which to study astrocyte biology and identifying Lapsyn as a key new molecule. It seems clear Lapsyn is required mng-autonomously for branch formation, early, rather than maintenance; it is a molecular pathway that distinguishes cortex glia from mngs; and suggests there is an interesting new system to study glial trophic support by neurons. They have also done a nice job of trying to tease apart the differences between FGF and Lapsyn phenotypes. Overall this is an exciting study that will advance our understanding of the molecular basis of astrocyte development.

Reviewer #2 (Remarks to the Author):

This study by Richier et al focuses on the identification and characterization of astrocyte like glia in the developing medulla. The study combines cutting edge techniques such as MARCM, flybow, protein traps etc to discover the origin and development of these astrocyte medulla neuropil glia. Furthermore, they perform an RNAi screen and identify Lapsyn, a relatively unstudied LRR protein, as required for normal branch extension of these glia as well as their migration, positioning and survival. Finally, the authors try to put Lapsyn in the context of the FGF signaling pathways, albeit with limited success.

Overall, I think this is an important tour de force that identifies a new glial type, characterizes its development as well as finds a transmembrane LRR protein that is required for its proper development. I do find the FGF part and the attempt to link Lapsyn to FGF the weakest spot of the paper and I propose to take it out, at least in its current state. I will offer a few alternatives to replace the FGF part but this is up to the authors. Finally, there are other issues that I think should be addressed - see specific comments.

1) The most important point is that I find that the FGF part (figure 7) and especially the linkage between Lapsyn and FGF (figure 8) unsatisfying. While I do see the logic in trying to address a potential link between Lapsyn and FGF, I think the end result is more confusing and really lacks any conclusive mechanistic understanding (hence the lack of a model). Specifically - genetic interactions are already a relatively weak mechanistic predictor - however, performing genetic interactions on a gain of function defect (UAS-Htl-lambda) is really not interpretable in my opinion. Therefore, I suggest that figure 8 (and ideally also 7) will be taken out of the manuscript - they don't provide any substantial addition but rather confuse the reader. Three options to expand the study instead of this direction: a) what is the phenotype of the Lapsyn mutant - the descriptive phenotype is very partial.

How about looking at cytoskeleton - actin, MTs, etc - all have relatively good UAS-reporters and can thus be combined in single cell MARCM analyses?; b) How do defective astrocytes (mngs) affect the neuronal circuit? With the multiple binary systems today, it is totally doable to manipulate one type of cell (glia) and look at another (neurons); c) Do other astrocytes require Lapsyn for their function? One easy place to look would be the astrocytes that are required for engulfing MB debris.

2) Fig 1: What is the proof that NP6520 and loco1.3D2 indeed label different glia? Obviously, there are population that one Gal4 labels but not the other. BUT - is there a real proof that the mngs labeled are not the same cells that are differentially labeled because of the strength of the G4? A textual clarification would suffice but if the authors really wanted to nail this then they would have to generate a Lexa or Q version of one of the lines and express both in the same animal to show that they are different cells.

3) Fig 1: Text describing k-h is not very decisive or informative. I think that tightening and focusing the text here would help. For example - there is a clear tiling - albeit with some minor overlaps - this can be said much clearer.

4) Fig 2- what is the purpose of panel b? Why is it called "control"? Figure S2 is actually more interesting and informative. The text describing 2c-d would benefit from some clarification - it took me a few reads to understand the logic. In 2g (and also elsewhere) - please show us where the inset comes from in the main figure.

5) Fig 3 - tubP>Gal80> is jargon - simplify and make accessible to non-Drosophila people. the phenotype in 3f-g is variable and described in a very fuzzy way... As it is quantified later - wouldn't it be nice to quantify here (with specific examples!)??

6) Fig4 - so I don't understand the partial rescue with Elav-Gal4? What does it mean? With this knowledge, how can the authors claim that Lapsyn is only required in glia?

7) Fig6 - the temporal organization of the figure is confusing. If the authors want to keep the order of the panels, then simple graphic modifications could help in separating these two parts.

8) Fig 7 - in addition to what I mentioned in #1; while protein traps are informative, one has to interpret the data with care - softening the text a bit is sufficient. Also - is b really wild type (as mentioned in the figure) or is this an animal that expresses the htl fosmid (as mentioned in the text)? and if the latter, why did you need to express htl rather than use a real wild type?

Taken together, this is an important paper that tackles a difficult and complicated glial cell. While the study is very broad, I think that the last part is confusing and therefore counterproductive.

Reviewer #3 (Remarks to the Author):

Overview This paper has considerable strengths and several novel aspects. First, the authors clarify existing descriptions of of glial cell types in the visual system, in particular the astrocyte-like mng cells which extend intricate processes among synapses within the medulla. These cells are particularly interesting because they show both columnar and layer-specific spatial domains, suggesting they could be an important model to understand the specificity with which glia and astrocytes interact at synapses. Second, to identify molecules that control mng development, they performed a genetic screen. Third, they then generated a number of new reagents to demonstrate the cell-autonomous function of the LRR protein Lapsyn in controlling the extension of branches within the mng arbors, and the positioning of mng. Lapsyn is largely uncharacterized, and resembles the NetrinG ligand found in vertebrates. Fourth, they characterized the relationship between Lapsyn and the FGF signaling pathway, which is also known to be essential for astrocyte shape.

Strengths – This is an interesting, well-written and comprehensive paper, employing a range of powerful techniques, that makes significant advances in understanding the cellular features that characterize mng in the optic lobe, and molecular mechanisms that act together to govern their shape,

position, and survival. Imaging is beautifully done, figures are well-wrought for the most part, and interpretations of the data are sound.

Limitations – The mechanism of Lapsyn function remains enigmatic, Lapsyn is insufficient to drive astrocyte-like morphogenesis when ectopically expressed in other glia, and whether Lapsyn is functionally conserved in other systems is unknown. However, given the breadth and quality of the paper, I see none of these limitations as a major weakness.

Minor points:

1. Further characterization of the molecular profile of mng could demonstrate how closely related these cells are to larval VNC astrocytes. For example, GAT, Eaat1, Prospero, Gs2.
2. The rescue assay could be used to demonstrate whether Lapsyn and NetrinG are functionally conserved.
3. Fig. 3c,d,e –are any subtypes of mng preferentially affected by loss of Lapsyn?
4. Fig 3g – are the notations “i, ii, and iii” necessary?
5. The description of the rescues in the main manuscript text seems clear, but to me the graphs in Fig. 4b are confusing, not at all intuitive, and do not seem to support the text. For clarity, the legend could be improved,, the separation between the left and right graphs enhanced, and the Y-axis labels reconsidered. Some of these genotypes appear to be missing from the list.
6. Fig. 5 I,j,k – the repo stain is impossible to see.
7. The reference list was omitted important contributions about the development and functions of astrocyte-like glia in the larval VNC and adult brain, including work from:
 - a. van Meyel lab Peco et al. Development. 2016 Apr 1;143(7):1170-81. and Stacey et al. J Neurosci. 2010 Oct 27;30(43):14446-57.
 - b. Birman lab Curr Biol. 2004 Apr 6;14(7):599-605.
 - c. Hidalgo lab Losada-Perez M et al. J Cell Biol. 2016 Aug 29;214(5):587-601
 - d. Jackson lab Ng et al. Curr Biol. 2011 Apr 26;21(8):625-34.

Point by point response to comments of reviewers

We would like to thank the reviewers for their positive assessment of our work and hope to have addressed the very helpful comments satisfactorily to strengthen our study.

Reviewer #1 (Remarks to the Author):

This is a very nice study from Salecker and colleagues that explores the development of medulla neuropil glia (mng). The authors describe the detailed morphology of these cells, developmental elaboration, cell-cell interactions with other mngs and neurons, and timing of infiltration relative to synaptogenesis. Most interestingly, though a forward genetic screen they identify Lapsyn, and LRR molecule required for mng growth and survival. They explore the timing of expression and function of Lapsyn, its roles in mng survival, and genetic interactions with FGF signaling. The authors should be commended for rigorous genetics, beautiful figures, and for establishing an interesting system in which to study astrocyte biology and identifying Lapsyn as a key new molecule. It seems clear Lapsyn is required mng-autonomously for branch formation, early, rather than maintenance; it is a molecular pathway that distinguishes cortex glia from mngs; and suggests there is an interesting new system to study glial trophic support by neurons. They have also done a nice job of trying to tease apart the differences between FGF and Lapsyn phenotypes. Overall this is an exciting study that will advance our understanding of the molecular basis of astrocyte development.

Reviewer #2 (Remarks to the Author):

This study by Richier et al focuses on the identification and characterization of astrocyte like glia in the developing medulla. The study combines cutting edge techniques such as MARCM, flybow, protein traps etc to discover the origin and development of these astrocyte medulla neuropil glia. Furthermore, they perform an RNAi screen and identify Lapsyn, a relatively unstudied LRR protein, as required for normal branch extension of these glia as well as their migration, positioning and survival. Finally, the authors try to put Lapsyn in the context of the FGF signaling pathways, albeit with limited success.

Overall, I think this is an important tour de force that identifies a new glial type, characterizes its development as well as finds a transmembrane LRR protein that is required for its proper development. I do find the FGF part and the attempt to link Lapsyn to FGF the weakest spot of the paper and I propose to take it out, at least in its current state. I will offer a few alternatives to replace the FGF part but this is up to the authors. Finally, there are other issues that I think should be addressed - see specific comments.

1) The most important point is that I find that the FGF part (figure 7) and especially the linkage between Lapsyn and FGF (figure 8) unsatisfying. While I do see the logic in trying to address a potential link between Lapsyn and FGF, I think the end result is more confusing and really lacks any conclusive mechanistic understanding (hence the lack of a model). Specifically - genetic interactions are already a relatively weak mechanistic predictor - however, performing genetic interactions on a gain of function defect (UAS-Htl-lambda) is really not interpretable in my opinion. Therefore, I suggest that figure 8 (and ideally also 7) will be taken out of the manuscript - they don't provide any substantial addition but rather confuse the reader.

The FGF signaling pathway is central to glial development across species with pleiotropic conserved effects on proliferation, migration and branch formation. Because the requirements of this pathway in astrocyte-like mng were not known, we believe that its assessment is essential to understand the context, in which *lapsyn* functions. If rescue experiments would have indicated that *lapsyn* functions downstream of this pathway, this part of our study would have been considered as straightforward. However, our data instead are in line with a more complex, but commonly observed scenario, in which two components work independently/in parallel, with shared and distinct roles. In our view, these findings are nevertheless significant.

Specifically, we demonstrate that the FGF signaling pathway plays a conserved role in branch

morphogenesis of astrocyte-like mng. Furthermore, we show that FGF signaling, as in other glial cells controls their proliferation and migration. However, in addition, we also provide evidence for a previously unrecognized function of this pathway in promoting glial cell survival. We had identified *htl* as one candidate in our original screen, and considering the ample knowledge of the conserved role of this pathway in glia, it seemed essential to us to explore its role in the visual system. Most importantly in our study, we explored the function of FGF receptor signaling to gain mechanistic insights into *lapsyn* function. Cell loss in the absence of *lapsyn* would have suggested that this LRR protein is directly involved in cell survival. However, our findings, assessing *lapsyn* function in the context of FGF pathway roles provide evidence that cell death is a consequence of incorrect positioning of mng in the cortex, while revealing an important gliotrophic role, likely of neurons, mediated by FGF signaling at the neuropil border. Furthermore, these findings highlight the robustness of astrocytes in controlling their branch morphogenesis through independent molecular mechanisms.

The reviewer questions the validity of one of our genetic approaches to test interactions of the pathway using the *UAS-htl-lambda* transgene. Generally, our interpretations are based on several genetic strategies, including a classic epistasis experiment, in which one component, the FGF receptor is knocked down, while attempting to rescue any arising defects by over-expressing a putative downstream factor, Lapsyn, in the same cell. Previously, the strategy of combining a loss-of-function allele with over-expression of the activated *htl* transgene has been successfully used in epistasis tests for Pyr and Ths (Stathopoulos et al., 2004), Dof (Michelson et al., 1998), Nesthocker (Mariappa et al., 2011), Pebble (Schumacher et al. 2004) and the Insulin receptor (Avet-Rochez et al., 2012) within different tissues, determining correctly upstream, downstream and in parallel relationships. Our detailed quantifications enabled us to conclude that *lapsyn* is not directly required for survival of mng that migrated ectopically into the cortex. In our view, there is currently no other conceivable way to generate surviving ectopic mng clones in the cortex than the over-expression of *htl^A* to test the role of *lapsyn* in controlling survival.

We therefore would like to argue that our findings are significant and the removal of Figures 7 and 8 would considerably weaken our ability to understand *lapsyn* function. We would also like to add that reviewers 1 and 3 did not perceive these findings negatively. To address this reviewer's concern for this part we modified the text to emphasize how the experiments on FGF signaling help us to understand the function of *lapsyn* and also added a model in Supplementary Figure 8.

Three options to expand the study instead of this direction:

We are grateful for these thoughtful suggestions. Carefully weighing the suggested experiments, we believe that the first and third options are most valuable and feasible to further strengthen our current study.

a) what is the phenotype of the Lapsyn mutant - the descriptive phenotype is very partial. How about looking at cytoskeleton - actin, MTs, etc - all have relatively good UAS-reporters and can thus be combined in single cell MARCM analyses?

Following this advice, we have examined the actin and microtubule cytoskeleton in wild type mng using fluorescent transgenic probes [*UAS-ChRFP-Tub* (microtubules), *UAS-Eb1-GFP* (tracking plus-end of microtubules), *UAS-nod-GFP* (for minus-end of microtubules), and *UAS-LifeAct-GFP* (actin)]. We observed that cytoskeletal elements in columnar astrocyte-like mng are differentially distributed: while microtubules are enriched in primary branches, actin filaments are found abundantly in secondary branches. Similar to the polarized organization of microtubules observed in VNC astrocytes (Stork et al., 2014), we observed Nod-GFP primarily in the cell body and Eb1-GFP in branches of mng (new Supplementary Fig. 3).

Assessing effects in MARCM clones would have required several generations of crosses to completely rebuild some stocks with a membrane-bound red fluorescent protein and actin- or microtubule green fluorescent protein reporters. However, the regular tiled columnar array of mng facilitates probing of the cytoskeletal organization even when visualizing the entire array of astrocyte-like mng. We therefore have assessed the cytoskeletal organization in mng, in which *lapsyn* has been knocked down. We observed that the distribution of the actin and microtubule cytoskeleton was disrupted following this genetic manipulation, highlighting a strong effect on actin-rich secondary branches (new Supplementary Fig. 3). Without knowing the pathway, in which *lapsyn* functions, it is

not possible to distinguish whether cytoskeletal abnormalities are caused directly by the loss of *lapsyn* or indirectly by abnormal branching or both. We therefore chose to present these data as Supplementary information.

b) How do defective astrocytes (mngs) affect the neuronal circuit? With the multiple binary systems today, it is totally doable to manipulate one type of cell (glia) and look at another (neurons);

This is a very important question, and we agree that in principle visualizing neurons while manipulating glia with the help of binary expression systems is feasible. However, we believe that addressing this question is the basis for a new substantial project beyond the scope of this study because of the complexity of the circuitry in the visual system.

The medulla is the most diverse region in the *Drosophila* brain consisting of approximately 40,000 neurons, corresponding to more than 70 different medulla neuron subtypes. These acquire their morphologies stepwise at different time points. Our data indicate that targeting of R-cell axons in the medulla is not affected following the loss of *lapsyn* in glia consistent with the fact that mng extend their branches into the neuropil after these neurons targeted to their correct layers. We also checked a subset of target neurons labeled with the reporter *isl-tau-myc-EGFP* and could not find any obvious projection defects (our unpublished observations).

Thus, a larger screen is needed to identify potentially affected neuron subtypes with the help of specific drivers that are active in adults and during development. Such *Gal4* or *LexA* based drivers are still generally rare. Moreover, for *LexA* drivers, new transgenes to knockdown *lapsyn* in mng will need to be generated. Importantly at this stage, it is not yet clear, whether astrocytes affect connectivity at the level of branching patterns or at the level of specific synaptic connections. Previous findings (Muthukumar et al. 2014) revealed that astrocytes are involved in promoting synapse formation in *Drosophila*. Similarly, we could assess the overall number of synapses using electron microscopy. However, without linking these findings to specific neuron subtypes, such data would likely not provide any truly new insights into astrocyte function during circuit assembly. Bringing such experiments to the next level would involve the visualization and quantification of synapses in specific neurons while manipulating glia, further underscoring that all these experiments constitute a new study.

c) Do other astrocytes require *Lapsyn* for their function? One easy place to look would be the astrocytes that are required for engulfing MB debris.

To strengthen the argument that *Lapsyn* plays a general role in astrocyte morphogenesis, we followed the valuable suggestion of this reviewer, as well as of reviewer 3. We demonstrate that the *loco*^{1.3D2}-*Gal4* driver is also active in astrocytes in the antennal lobe, mushroom body, central brain and VNC. Co-labeling of flies expressing the *lapsyn* fosmid with Bruchpilot (Brp) or Fas2 antibodies revealed that *Lapsyn* protein is expressed widely in astrocytic branches in addition to other CNS glia. Knockdown in these cells and analysis in the adult revealed that *lapsyn* is required in all astrocytes examined to mediate branch morphogenesis. Because astrocytic branches in adult mushroom body lobes were unexpectedly sparse compared to other CNS region, effects on branching were less clear to assess in this neuropil. Our data suggest that mushroom body remodeling was not affected following *lapsyn* knockdown (see enclosed reviewer Fig. below). This observation would be in line with the notion that *lapsyn* mediates branch extension during late pupal development but not the earlier very different steps required for pruning during early metamorphosis. However, because of the low level of resolution, we would prefer to not include these panels into our manuscript.

Reviewer Fig. Richier et al. rev

Reviewer Fig. Analysis of effects of *lapsyn* knockdown on mushroom body remodeling. Fas2 was used to label the mushroom body α , β and γ lobes (red, **a,b**). GFP labeling is not shown. Compared to controls (**a**, n=9) *lapsyn* knockdown (**b**, n=9) in astrocyte-like glia does not appear to affect remodeling of mushroom body neurons at this level of resolution (**b**). Panels **a,b** show projections of optical sections. Scale bars, 50 μ m.

2) Fig 1: What is the proof that NP6520 and *loco1.3D2* indeed label different glia? Obviously, there are populations that one Gal4 labels but not the other. BUT - is there a real proof that the mngs labeled are not the same cells that are differentially labeled because of the strength of the G4? A textual clarification would suffice but if the authors really wanted to nail this then they would have to generate a Lexa or Q version of one of the lines and express both in the same animal to show that they are different cells.

The distinction was primarily based on the characteristic adult morphologies of mng labeled by these two drivers and available data in the literature describing *NP6520-Gal4* as being active in ensheathing glia in other parts of the nervous system (Awasaki et al., 2008). Another driver known to be active in ensheathing glia, *Mz0709-Gal4*, labeled similarly sparsely branched mng (in addition to some astrocyte-like mng, our unpublished observations). By contrast, *loco^{1.3D2}-Gal4* and *alrm-Gal4* label abundantly branched astrocyte-like mng, arguing for the existence of two different mng populations. However, as the reviewer very helpfully pointed out, it is not possible to completely exclude differences in labeling strength. To further strengthen this conclusion, we turned to another recently published driver, *R56F03-Gal4* (Peco et al. 2016;). Comparing *NP6520-Gal4* and *R56F03-Gal4* activity, we observed similar expression patterns. However, because the *NP6520-Gal4* driver proved to be more variable, we switched to the new *R56F03-Gal4* driver. Extending above recent descriptions (Kremer et al., 2017), we combined this driver with *FB1.1B^{260b}*, *loco^{1.3}-LexA* and *lexAop-cd8mCherry* transgenes. Assessing immunolabeled samples with Repo and Prospero, we observed distinct expression in closely associated sparsely branched Prospero-negative ensheathing mng and in abundantly branched Prospero-positive astrocyte-like mng in the adult medulla. Together these findings confirm that ensheathing and astrocyte-like mng are indeed separate populations of glia. These new findings are shown in a modified Figure 1 and Supplementary Fig. 1.

3) Fig 1: Text describing k-h is not very decisive or informative. I think that tightening and focusing the text here would help. For example - there is a clear tiling - albeit with some minor overlaps - this can be said much clearer.

We attempted to simplify the text for these panels both in the main text and the Figure legend. Reducing the findings to tiling, however, would be too limited as conclusion, as this would omit the interesting differential distribution of glial and neuronal processes within synaptic medulla neuropil layers, that will likely have functional implications.

4) Fig 2- what is the purpose of panel b? Why is it called "control"? Figure S2 is actually more interesting and informative.

The purpose of panel 2b is to document the migratory path of future mng through the cortex to the most anterior and youngest border of the medulla neuropil, which is best illustrated in a projection of several optical sections. In the revised manuscript, we explain this in more detail. Following the suggestion of the reviewer, we have moved the panels of the original Supplementary Fig. 2 to the

main Fig. 2. We agree that describing this panel as control is misleading, and removed it. The complete genotype is provided in Supplementary Table 1.

The text describing 2c-d would benefit from some clarification - it took me a few reads to understand the logic.

We revised the related text in the Results and Figure legend to facilitate the understanding.

In 2g (and also elsewhere) - please show us where the inset comes from in the main figure.

As suggested, we added small boxes in panels 2c and g, as well as in all other Figures to delineate the areas from which insets were taken.

5) Fig 3 - *tubP>Gal80>* is jargon - simplify and make accessible to non-Drosophila people.

We had provided the detailed explanation of the screen strategy and the function of the *tub>Gal80>* transgene in the Methods section, but agree that it is important to provide this information in the main section of the manuscript and therefore provided an explanation in the legend of Fig. 3 to facilitate the understanding of our approach.

the phenotype in 3f-g is variable and described in a very fuzzy way... As it is quantified later - wouldn't it be nice to quantify here (with specific examples!)??

Defects caused by RNAi-mediated knockdown of *lapsyn* are qualitatively similar to our findings using the MARCM approach. Defects generated by RNAi-mediated knockdown generally tend to be more variable and less strong. Moreover in knockdown experiments, phenotypes would be assessed in a genetic background, in which *lapsyn* is affected in all mng, while in MARCM clones only individual mng in a largely heterozygous background would be affected. In our view, quantifications would thus be less meaningful for interpretations compared to MARCM clones, and were therefore not provided to not duplicate findings. However in the revised manuscript, we included the suggested information regarding the distribution of defects as percentages to strengthen the data obtained by RNAi-mediated knockdown. The chosen samples were already representative examples of phenotypes.

6) Fig4 - so I don't understand the partial rescue with *Elav-Gal4*? What does it mean? With this knowledge, how can the authors claim that *Lapsyn* is only required in glia?

The observation that expression of *Lapsyn* using *elav-Gal4* is able to rescue lethality during larval stages but not in adults is in line with the original observations of Guan et al. 2011. We have assessed *Lapsyn* expression in second and third instar larval VNCs using the *lapsyn*^{TRG027706-135B} line, and solely detected expression in glia. *elav-Gal4*^{c155} is known to be transiently active in glia during early development (Berger et al. 2007). These low levels may enable very unhealthy animals to survive during early larval development. We have added this possible explanation to the text in the Method sections.

7) Fig6 - the temporal organization of the figure is confusing. If the authors want to keep the order of the panels, then simple graphic modifications could help in separating these two parts.

We agree that the chosen order starting with older stages to assess branching defects followed by younger stages to assess cell loss and positioning defects may not be understandable when just looking at the figure. We therefore spaced panels a-f and g-l more prominently, and added titles on the left-hand titles (mng branch extension, mng positioning).

8) Fig 7 - in addition to what I mentioned in #1; while protein traps are informative, one has to interpret the data with care - softening the text a bit is sufficient.

We agree that expression patterns of fosmid as new reagents need to be interpreted carefully. We therefore modified the text to indicate that the *htl*^{TRG482} reports strong expression of this FGF receptor. We had conducted stainings with an antibody directed against Htl in third instar larval optic lobes and had detected expression in mng in line with the expression reported by the fosmid. However, the quality of staining was low, and we therefore chose to not include these data in our manuscript.

Also - is b really wild type (as mentioned in the figure) or is this an animal that expresses the *htl* fosmid

(as mentioned in the text)? and if the latter, why did you need to express htl rather than use a real wild type?

The genotype of pupae, whose optic lobes were stained with an antibody against Dof in Fig. 7b is indeed *wild type*^{OreR} (see also Supplementary Table 1). We modified the text in the figure legend to state this more clearly.

Taken together, this is an important paper that tackles a difficult and complicated glial cell. While the study is very broad, I think that the last part is confusing and therefore counterproductive.

Reviewer #3 (Remarks to the Author):

Overview This paper has considerable strengths and several novel aspects. First, the authors clarify existing descriptions of glial cell types in the visual system, in particular the astrocyte-like mng cells which extend intricate processes among synapses within the medulla. These cells are particularly interesting because they show both columnar and layer-specific spatial domains, suggesting they could be an important model to understand the specificity with which glia and astrocytes interact at synapses. Second, to identify molecules that control mng development, they performed a genetic screen. Third, they then generated a number of new reagents to demonstrate the cell-autonomous function of the LRR protein Lapsyn in controlling the extension of branches within the mng arbors, and the positioning of mng. Lapsyn is largely uncharacterized, and resembles the NetrinG ligand found in vertebrates. Fourth, they characterized the relationship between Lapsyn and the FGF signaling pathway, which is also known to be essential for astrocyte shape.

Strengths – This is an interesting, well-written and comprehensive paper, employing a range of powerful techniques, that makes significant advances in understanding the cellular features that characterize mng in the optic lobe, and molecular mechanisms that act together to govern their shape, position, and survival. Imaging is beautifully done, figures are well-wrought for the most part, and interpretations of the data are sound.

Limitations – The mechanism of Lapsyn function remains enigmatic, Lapsyn is insufficient to drive astrocyte-like morphogenesis when ectopically expressed in other glia, and whether Lapsyn is functionally conserved in other systems is unknown. However, given the breadth and quality of the paper, I see none of these limitations as a major weakness.

Our study consists of a detailed characterization of astrocyte mng from birth to adulthood, a screen identifying a new molecule controlling astrocyte branch morphogenesis and anchorage, the functional expression of phenotypes and its relation to the well-known FGF pathway controlling glial development in other regions of the brain. In an attempt to overcome the above-mentioned limitations, we have now included new data demonstrating that *lapsyn* is expressed in and required for branch extension not only in astrocytes in the visual system, but also in the antennal lobe, the central brain and adult and larval VNC, underscoring its wider central role. We also have assessed the impact on the actin and microtubule cytoskeleton. To gain insights into the mechanisms underlying Lapsyn function, it will be essential to identify and characterize the binding partners of this LRR protein in cis and in trans. Because LRR proteins can bind to a large variety of molecule classes, and some of these will be likely provided by neurons, new unbiased approaches will be required. We strongly believe that this constitutes a substantial new study.

Minor points:

1. Further characterization of the molecular profile of mng could demonstrate how closely related these cells are to larval VNC astrocytes. For example, GAT, Eaat1, Prospero, Gs2.

In the revised version of our manuscript, we have now included images illustrating the expression pattern of above markers. These findings highlight significant similarities between astrocyte-like mng and other astrocyte-like glial subtypes in the *Drosophila* CNS. These data have been included in a new Supplementary Fig. 2.

2. The rescue assay could be used to demonstrate whether Lapsyn and NetrinG are functionally conserved.

Data base searches identified the NetrinG ligand family as potential remote homologs of Lapsyn because of similarities in their LRR composition. The sequence homology of 29% between *Drosophila* and mouse/human proteins is relatively low. Moreover, proteins differ in their overall domain composition, as vertebrate NGL1-3 family members contain an additional extracellular immunoglobulin domain and an intracellular PDZ [Postsynaptic density protein (PSD95), Discs large tumor suppressor (Dlg1), and Zonula occludens-1 protein (zo-1)] binding motif. Individual sequences even among vertebrate family members vary. The suggested rescue experiment would thus involve the generation of three new transgenic lines to express for instance mouse NGL1-3 family members with the help of the *Gal4/UAS* system in *mng*. Altogether, this would require at least 7 generations of crosses. Furthermore, constructs would need to be tested for expression levels and over-expression defects. Considering that NGL1-3 and Lapsyn are distantly related rather than orthologs, we strongly believe that the proposed lengthy experiment would not allow us to gain insights, which would be easily interpretable with our current knowledge. We modified the text to better highlight the distant relationship between NGL1-3 and Lapsyn in the main text.

3. Fig. 3c,d,e –are any subtypes of *mng* preferentially affected by loss of Lapsyn?

Assessing optic lobes, in which *lapsyn* was knocked down (Fig. 3) or lost (Fig. 5), we did not observe any differences in branching defects in *dmng*, *pmng* or *lmng*. We altered the wording to describe this in the result section.

4. Fig 3g – are the notations “i, ii, and iii” necessary?

We have removed i, ii, and iii.

5. The description of the rescues in the main manuscript text seems clear, but to me the graphs in Fig. 4b are confusing, not at all intuitive, and do not seem to support the text. For clarity, the legend could be improved, the separation between the left and right graphs enhanced, and the Y-axis labels reconsidered. Some of these genotypes appear to be missing from the list.

We appreciate this suggestion and have revised the legend, separated the two graphs, modified the labels, and also added a more comprehensive description of genotypes in the Method section (thank you for alerting us to this).

6. Fig. 5 l,j,k – the repo stain is impossible to see.

Repo labeling is shown in dark blue in these panels, which is sometimes hard to see when printed. We therefore changed the shade to a brighter blue in the revised figure.

7. The reference list was omitted important contributions about the development and functions of astrocyte-like glia in the larval VNC and adult brain, including work from:

- a. van Meyel lab Peco et al. *Development*. 2016 Apr 1;143(7):1170-81. and Stacey et al. *J Neurosci*. 2010 Oct 27;30(43):14446-57.
- b. Birman lab *Curr Biol*. 2004 Apr 6;14(7):599-605.
- c. Hidalgo lab Losada-Perez M et al. *J Cell Biol*. 2016 Aug 29;214(5):587-601
- d. Jackson lab Ng et al. *Curr Biol*. 2011 Apr 26;21(8):625-34.

We were aware of these important contributions, but were not able to add these studies because of the limitation to 70 references. We now have added some of these references but had to find a compromise by removing some references and moving others related to the genetic tools used into a Supplementary Table 3 and References.

REVIEWERS' COMMENTS:

Reviewer #2 (Remarks to the Author):

In this revised manuscript the authors have addressed the large bulk of my concerns. The addition of the cytoskeletal characterization and Lapsyn involvement in astrocyte biology outside the eye are good addition. I therefore think that the overall paper is strong and an important addition to our understanding of glial biology and should be accepted without additional changes.

That said, I still find figure 8 the weakest part of this paper - as in my opinion it does not come up with a clear proposed model. I do not see the direct evidence that lapsyn is required for mng anchorage. The authors should consider if and how they want to maintain this figure in the manuscript. all in all, this is a minor point for the paper and should not affect its publication.

Reviewer #3 (Remarks to the Author):

The authors have responded to the initial reviews with added experiments, thoughtful responses, and appropriate revisions.

This is a strong, thorough, and novel body of work. The experiments are convincing, and I have no doubt these are reproducible findings that will influence thinking in the field because they provide molecular evidence for a new mechanism influencing the growth and morphology of astrocytes.

Point by point response to comments of reviewers

We would like to thank the reviewers for their positive evaluation of our study.

Reviewer #2 (Remarks to the Author):

In this revised manuscript the authors have addressed the large bulk of my concerns. The addition of the cytoskeletal characterization and Lapsyn involvement in astrocyte biology outside the eye are good addition. I therefore think that the overall paper is strong and an important addition to our understanding of glial biology and should be accepted without additional changes.

That said, I still find figure 8 the weakest part of this paper - as in my opinion it does not come up with a clear proposed model. I do not see the direct evidence that lapsyn is required for mng anchorage. The authors should consider if and how they want to maintain this figure in the manuscript. all in all, this is a minor point for the paper and should not affect its publication.

After careful consideration, we would prefer to keep the current version of Fig. 8 in the main section of our manuscript. The experiments presented in this figure assess *lapsyn* function in the context of the FGF pathway. These data allow us to show that FGF signaling and *lapsyn* are not simply epistatic to each other and function in parallel to promote branch morphogenesis. Importantly, our findings support the notion that *lapsyn* does not play a direct a role in mediating survival. To provide additional evidence for a direct role of *lapsyn* in positioning at the neuropil border, we had co-expressed *lapsyn* and the constitutive-active Heartless receptor in astrocyte-like mng and observed a reduction of these cells in the cortex (Supplementary Fig. 7). Future studies identifying the interacting partners of *lapsyn* will be instrumental to provide insights into the underlying mechanism.

Reviewer #3 (Remarks to the Author):

The authors have responded to the initial reviews with added experiments, thoughtful responses, and appropriate revisions.

This is a strong, thorough, and novel body of work. The experiments are convincing, and I have no doubt these are reproducible findings that will influence thinking in the field because they provide molecular evidence for a new mechanism influencing the growth and morphology of astrocytes.